# Effects of Far-Red Light and Ultraviolet Light-A on Growth, Photosynthesis, Transcriptome, and Metabolome of Mint (*Mentha haplocalyx* Briq.)

**DOI:** 10.3390/plants13243495

**Published:** 2024-12-14

**Authors:** Lishu Yu, Lijun Bu, Dandan Li, Kaili Zhu, Yongxue Zhang, Shaofang Wu, Liying Chang, Xiaotao Ding, Yuping Jiang

**Affiliations:** 1College of Ecological Technology and Engineering, Shanghai Institute of Technology, Shanghai 201418, China; 15921457938@163.com (L.Y.); lddwyzh2022@163.com (D.L.); z18253926832@163.com (K.Z.); 2Shanghai Key Laboratory of Protected Horticultural Technology, Horticultural Research Institute, Shanghai Academy of Agricultural Sciences, Shanghai 201403, China; xuezylemon@foxmail.com (Y.Z.); sfwu@saas.sh.cn (S.W.); 3Shanghai Sunqiaoyijia Tech-Agriculture Co., Ltd., Shanghai 201210, China; lijunbu@126.com; 4School of Agriculture and Biology, Shanghai Jiao Tong University, Shanghai 200240, China; changly@sjtu.edu.cn

**Keywords:** mint, far-red radiation, ultraviolet-a, transcriptome, metabolome, phenylpropanoid, flavonoid

## Abstract

To investigate the effects of different light qualities on the growth, photosynthesis, transcriptome, and metabolome of mint, three treatments were designed: (1) 7R3B (70% red light and 30% blue light, CK); (2) 7R3B+ far-red light (FR); (3) 7R3B+ ultraviolet light A (UVA). The results showed that supplemental FR significantly promoted the growth and photosynthesis of mint, as evidenced by the increase in plant height, plant width, biomass, effective quantum yield of PSII photochemistry (F_v_’/F_m_’), maximal quantum yield of PSII (F_v_/F_m_), and performance index (PI). UVA and CK exhibited minimal differences. Transcriptomic and metabolomic analysis indicated that a total of 788 differentially expressed genes (DEGs) and 2291 differential accumulated metabolites (DAMs) were identified under FR treatment, mainly related to plant hormone signal transduction, phenylpropanoid biosynthesis, and flavonoid biosynthesis. FR also promoted the accumulation of phenylalanine, sinapyl alcohol, methylchavicol, and anethole in the phenylpropanoid biosynthesis pathway, and increased the levels of luteolin and leucocyanidin in the flavonoid biosynthesis pathway, which may perhaps be applied in practical production to promote the natural antibacterial and antioxidant properties of mint. An appropriate increase in FR radiation might alter transcript reprogramming and redirect metabolic flux in mint, subsequently regulating its growth and secondary metabolism. Our study uncovered the regulation of FR and UVA treatments on mint in terms of growth, physiology, transcriptome, and metabolome, providing reference for the cultivation of mint and other horticultural plants.

## 1. Introduction

In recent years, artificial light has been extensively utilized in horticultural facilities, and studying its effects on the growth and development of garden plants is useful for enhancing productivity and quality. A plant factory is a controlled environment facility for plant production, utilizing light emitting diodes (LEDs) as the primary aluminum source, which has numerous benefits such as low energy use, extended lifespan, and narrow spectral range [1]. The combination of red and blue light (RB) is the predominant spectrum used in plant illumination, as red and blue wavelengths are the primary drivers of photosynthesis in plants [2]. Recent research on tomatoes demonstrated that the combination of RB light significantly increased the net photosynthetic rate (P_n_) and carbohydrate accumulation in comparison to white light [3]. The impacts of ultraviolet (UV) and far-red (FR) light on plants have been researched as well. UV light positively influences plant dwarfing [4], branching [5], and defense mechanisms [6]. In contrast, FR light regulates the state of photosensitive pigments, plant morphology, and photosynthetic capacity while promoting biomass accumulation [7]. Kong et al. discovered that supplementary FR light enhanced the stem length of grapes and increased carbohydrate levels in several organs and the maximum net photosynthetic rate in leaves [8]. Three types of UV differentially promote the expression of shikimate pathway genes and the production of anthocyanins in grape berries [9]. Ultraviolet-A (UVA) radiation significantly increased the shoot dry weight, leaf area, anthocyanin, and ascorbic acid levels of lettuce [10]. Yang et al. found that, compared with treatment without FR, adding a low or high proportion of FR can increase plant height, biomass, leaf area, and starch content of soybeans [11]. Field supplementation of ultraviolet-B radiation (UVB) reduced plant height, leaf area, and biomass of soybeans, alongside P_n_, transpiration rate (T_r_), and water use efficiency (WUE), while enhancing stomatal resistance. The reduction in chlorophyll (Chl) *b* content exceeded that of Chl *a* after field supplementation with UVB, suggesting that UVB predominantly impairs the light-harvesting pigments in soybeans, resulting in diminished light absorption and conversion efficiency by chloroplasts [12]. The findings indicated that various plants exhibit distinct responses to different light wavelengths and that adjusting spectral compositions can enhance plant growth. The addition of FR and UVA to pure blue light had a slight effect on plant development and morphology, exhibiting differing sensitivities across species [13].

LED light quality also had a dramatic effect on the accumulation of metabolites in plants [14]. In contrast to white light, red or blue LEDs promoted the accumulation of primary and secondary metabolites, including soluble sugars, starch, vitamin C, proteins, and polyphenols [15]. LEDs also influence plant hormone levels. Ding et al. found that artificial light improves the performance of grafted tomato seedlings throughout the recovery period by regulating endogenous hormone levels. In contrast to shading treatment, the levels of salicylic acid (SA) and kinetin (Kin) in grafted tomato seedlings treated with artificial light restoration significantly increased, while the content of indole-3-acetic acid (IAA) and jasmonic acid (JA) significantly decreased [16].

Current research on the impact of LEDs on plants mostly concentrates on crops including vegetables and fruits, researching yield, physiological morphology, biochemical, and related factors. Indeed, plants respond through specific photomorphogenic and physiological processes, at both the micro and macro level [17], such as through the improvement of growth and photosynthetic capacity in basil [18], flowering and carbohydrate accumulation of ageratum and salvia [19], and secondary metabolite content of *Myrtus communis* L [20]. Limited research has been undertaken on the effect of the LED spectrum on mint, a significant fragrant crop. Mint (*Mentha haplocalyx* Briq.), a perennial herbaceous species in the Lamiaceae family, offers both aesthetic appeal and fragrant qualities, presenting significant opportunities for development and promotion. Mint serves as a dual-purpose plant for medicinal and culinary applications, encompassing many bioactive components, such as phenols, flavonoids, terpenes, and organic acids, which exhibit antioxidant and antibacterial effects [21]. It is extensively employed in medicine, cosmetics, and the food industry [22]. With the enhancement of living standards and a heightened focus on quality of life, fragrant crops have become essential in landscaping. Numerous plants in the Lamiaceae family possess fragrant qualities, with mint being a prominent representative. Recent studies on mint have concentrated on its pharmacology, propagation, plant management, volatile ingredient extraction, and component analysis [23]. The influence of different light qualities on mint is predominantly restricted to monochromatic red, blue, or RB lights. Nevertheless, comparatively few studies have been undertaken regarding the impacts of FR and UVA light on the growth, photosynthesis, and metabolism of mint. This study aimed to study the impact of applying FR and UVA light on the growth, morphology, photosynthesis, transcriptome, and metabolome of mint, based on a fixed ratio of RB light, and to provide reference data for the cultivation of mint and other horticultural species.

## 2. Results

### 2.1. Effects of Different Light Qualities on Mint Growth Parameters

The morphology of mint showed significant differences under different light qualities. Mint plants under 70% red light and 30% blue light (7R3B) + FR (FR) treatment exhibited greater growth, as evidenced by higher plant height and larger shoots compared to the other two treatments. Plant growth under both 7R3B (CK) and 7R3B + UVA (UVA) treatments were comparatively uniform (Figure 1). In order to dynamically monitor the impact of different light qualities on the growth of mint plants, we regularly measured plant height and plant width. Within 7 days of treatment, no significant differences in plant height were observed among the three groups. From day 7 post-treatment, the plant height of mint under FR treatment increased rapidly and remained higher than the other two treatments throughout. The plant height of mint under CK and UVA treatments exhibited similar trends over time (Figure 2A). From day 24 post-treatment, the overall trends in plant width among three treatments were FR > CK > UVA (Figure 2B). On the 34th day of treatment, the plant height under FR treatment was 45.50% higher than those under CK, while UVA made no difference on plant height (Figure 2C). Compared with CK, the plant width increased by 13.07% under FR treatment, but there was no significant difference between UVA and CK (Figure 2D).

The biomass of mint under different light treatments was measured. In comparison to CK, FR treatment increased shoot fresh weight by 39.83% (Figure 3A), root fresh weight by 37.30% (Figure 3B), shoot dry weight by 35.80% (Figure 3C), and root dry weight by 37.71% (Figure 3D). Conversely, UVA treatment did not significantly affect any of these parameters.

### 2.2. Effects of Different Light Qualities on Mint Chlorophyll and Carotenoid Contents

In comparison to CK, the Chl *a*, Chl *b*, and total Chl contents of mint under FR treatment exhibited significant reductions of 9.94%, 9.26%, and 9.77%, respectively. UVA treatment did not significantly affect the Chl content in mint. Significant differences in carotenoid (Car) content were observed among the three treatments, with a trend indicating CK > UVA > FR (Table 1).

### 2.3. Effects of Different Light Qualities on Mint Photosynthesis

With the increase in irradiance, the P_n_ of all treatments exhibited varying degrees of enhancement. P_n_ displayed sensitivity to irradiance at levels below 400 μmol m^−2^ s^−1^, demonstrating a rapid increase in P_n_ as irradiance increased. When irradiance > 400 μmol m^−2^ s^−1^, P_n_ showed a gradual increase with rising irradiance, ultimately stabilizing. The trend of P_n_ among the three treatments was observed as follows: FR > CK > UVA (Figure 4).

In contrast with CK treatment, there was no significant difference in the P_n_ of FR and UVA treatments, but the trend of P_n_ under the three treatments was shown as: FR > CK > UVA (Figure 5A). The intercellular CO_2_ concentration (C_i_) of mint revealed a significant reduction of 6.38% under FR treatment compared to CK, while an increase of 7.33% was observed under UVA treatment (Figure 5B). Compared to CK treatment, the stomatal conductance (G_s_) of mint decreased by 26.00% and 28.09% under FR and UVA treatments, respectively (Figure 5C). The T_r_ was notably elevated in the CK treatment than in the other two treatments, implying a trend of CK > FR > UVA (Figure 5D). The WUE increased by 29.43% and 39.97% under FR and UVA treatments, respectively, in comparison to CK (Figure 5E).

In addition, actual photochemical efficiency of PSII (ΦPSII), electron transport rate (ETR) and photochemical quenching coefficient (qP) exhibited decreases of 15.95%, 15.77% and 15.13%, respectively, when treated with UVA in comparison to CK. However, FR treatment did not produce a significant effect on the three chlorophyll fluorescence parameters (Figure 6A–C). In contrast, effective quantum yield of PSII photochemistry (F_v_’/F_m_’), maximal quantum yield of PSII (F_v_/F_m_) and performance index (PI) of mint leaves increased under FR treatment, while no significant difference was observed between UVA and CK treatments (Figure 6D–F).

### 2.4. Effects of Different Light Qualities on Mint Transcriptome

#### 2.4.1. Differentially Expressed Genes (DEGs) Analysis of Mint Under Different Light Qualities

Principal component analysis (PCA) demonstrated the significance of samples and the overarching trend of variation among different treatments. The samples of CK, FR, and UVA exhibited strong repeatability, making them suitable for subsequent differential analysis (Figure 7A). DEGs were identified in CK, FR, and UVA samples based on the criteria of Fold Change ≥ 2 and False Discovery Rate (FDR) < 0.05. A total of 788 DEGs were identified in the comparison of FR and CK, comprising 532 up-regulated DEGs and 256 down-regulated DEGs. A total of 854 DEGs were identified in contrast to UVA and CK, involving 601 up-regulated and 253 down-regulated genes (Figure 7B). A Venn diagram was employed to illustrate the quantities of shared and unique DEGs among the different comparison groups. There were 359 DEGs common to both the FR vs. CK and UVA vs. CK groups, while 429 DEGs were unique to the FR vs. CK group, and 495 DEGs were unique to the UVA vs. CK group (Figure 7C).

#### 2.4.2. Kyoto Encyclopedia of Genes and Genomes (KEGG) Pathway Enrichment Analysis of DEGs in Mint Under Different Light Qualities

Pathway enrichment analysis utilized the KEGG Pathway Database of DEGs. Figure 8A,B illustrated the top twenty pathways which were enriched with DEGs. DEGs in FR vs. CK and UVA vs. CK were enriched in pentose and glucuronate interconversions. In addition, DEGs in UVA vs. CK were also enriched in phenylpropanoid biosynthesis, as well as starch and sucrose metabolism. The results demonstrated that FR and UVA light showed intricate influences on biological pathways in mint.

### 2.5. Effects of Different Light Qualities on Mint Metabolome

#### 2.5.1. Differential Accumulated Metabolites (DAMs) Analysis of Mint Under Different Light Qualities

The raw data collected using MassLynx V4.2 is processed by Progenesis QI V2.5 for peak extraction, peak alignment and other data processing operations, based on the Progenesis QI V2.5 online METLIN database and Biomark’s self-built library for identification and, at the same time, theoretical fragment identification and parent ion mass deviation all are within 100 ppm. Metabolomic qualitative and quantitative analyses were performed on nine samples using the LC-QTOF platform, resulting in the annotation of 4169 metabolites. Metabolites with Fold Change ≥ 1, VIP ≥ 1, and a *p*-value < 0.05 were defined as DAMs. A total of 2291 DAMs were identified in the FR vs. CK group, comprising 1150 up-regulated and 1141 down-regulated metabolites. In the UVA vs. CK group, a total of 2267 DAMs with 911 up-regulated and 1356 down-regulated metabolites were identified (Table 2). A Venn diagram illustrated that 1742 DAMs were common to both the FR vs. CK and UVA vs. CK groups, while 549 and 525 unique DAMs were contained, respectively (Figure 9).

#### 2.5.2. KEGG Pathway Enrichment Analysis of DAMs in Mint Under Different Light Qualities

Metabolites interact within organisms to form various pathways. In this study, all identified metabolites were annotated with reference to the KEGG database. We obtained the compound ID for KEGG KO from the KEGG pathway official website (https://www.kegg.jp/). The top twenty annotations with the highest number of KEGG ORTHOLOGY (KO) pathway level three entries were selected. The metabolites associated with the biosynthesis of other secondary metabolites were the most abundant, encompassing isoquinoline alkaloid biosynthesis, biosynthesis of various plant secondary metabolism, flavonoid biosynthesis, phenylpropanoid biosynthesis, and flavone and flavonol biosynthesis (Figure 10A). The ten most significantly up-regulated and down-regulated DAMs in the experimental group compared to the CK group were identified. The results demonstrated that the up-regulated DAMs were predominantly linked to nucleotide, flavonoid, and glucosinolate in FR vs. CK and UVA vs. CK groups (Figure 10B,C). The KEGG pathways of DAMs in the FR vs. CK group were enriched in pyruvate metabolism, monobactam biosynthesis, and isoquinoline alkaloid biosynthesis. Furthermore, DAMs in the UVA vs. CK group exhibited enrichment in porphyrin metabolism and glutathione metabolism (Figure 10D,E).

### 2.6. Combined Analysis of Transcriptome and Metabolome in Mint Under Different Light Qualities

#### 2.6.1. The Effects of Different Light Qualities on the Hormone-Related Pathways in Mint

The study identified that pathways related to growth, including plant hormone signal transduction, pentose and glucuronate interconversions, and starch and sucrose metabolism, displayed enrichment under both FR vs. CK and UVA vs. CK groups. Plant hormone signal transduction pathway was notably enriched in the two groups. We selected several hormone signal-related genes and analyzed their expression patterns in response to different light qualities, along with the accumulation of hormones. In the IAA signaling pathway, the expression of *the auxin influx carrier* (*AUX1*), *auxin response factor* (*ARF*), and *the auxin-responsive GH3 gene family* (*GH3*) were up-regulated in both FR and UVA treatments. The expression of *auxin-responsive protein IAA* (*AUX/IAA*) was up-regulated in FR treatment but down-regulated in UVA treatment. In the gibberellin (GA) signaling pathway, the expression of *the gibberellin receptor GID1* (*GID1*) was increased by both FR and UVA treatments. In contrast, the expression of *DELLA protein* (*DELLA*) was exclusively up-regulated in response to UVA treatment. In the abscisic acid (ABA) signaling pathway, FR treatment resulted in the up-regulation of *protein phosphatase 2C* (*PP2C*) expression. In the ethylene (ET), brassinosteroid (BR), and JA signaling pathways, the expression of *mitogen-activated protein kinase kinase 4/5* (*SIMKK*), *BRI1 kinase inhibitor 1* (*BKI1*), *xyloglucan:xyloglucosyl transferase TCH4* (*TCH4*), and *jasmonate ZIM domain-containing protein* (*JAZ*) was up-regulated in FR and UVA treatments. In the SA signaling pathway, the level of *transcription factor TGA* (*TGA*) increased under FR treatment. This analysis determined that variations in light qualities significantly influence the accumulation and distribution of plant hormones in mint leaves, regulating the developmental and morphological aspects of seedling growth (Figure 11).

#### 2.6.2. The Effects of Different Light Qualities on the Secondary Metabolism-Related Pathways in Mint

To investigate the impact of adding FR and UVA light with RB light on genes and metabolites associated with the secondary metabolism of mint, we selected pathways that were significantly enriched in both FR vs. CK and UVA vs. CK groups. The interaction of DEGs and DAMs with phenylpropanoid and flavonoid biosynthesis pathways was analyzed.

In relation to phenylpropanoid biosynthesis, the expressions of *phenylalanine ammonia-lyase* (*PAL*), *cinnamyl-alcohol dehydrogenase* (*CAD*), and *peroxidase* (*POD*) were found to be up-regulated, while the expression of *coniferyl-aldehyde dehydrogenase* (*REF1*) was down-regulated under both FR and UVA treatments. *Scopoletin glucosyltransferase* (*TOGT1*) and *coniferyl-alcohol glucosyltransferase* (*UGT72E*) exhibited up-regulation in response to UVA exposure. Downstream metabolites, including methylchavicol, anethole, coniferin, and sinapyl alcohol, exhibited up-regulation in FR. Anethole, coniferin, and syringin were down-regulated in UVA (Figure 12).

The expression of *phlorizin synthase* (*PGT1*) and *flavonoid 3’,5’-hydroxylase* (*F3′5′H*) was up-regulated in FR and UVA treatments among the genes associated with flavonoid biosynthesis. *Chalcone synthase* (*CHS*) expression increased in FR, while *flavanone 7-O-glucoside 2’’-O-beta-L-rhamnosyltransferase* (*C12RT1*) and *bifunctional dihydroflavonol 4-reductase/flavanone 4-reductase* (*DFR*) expressions increased in UVA. Among the downstream metabolites, pinocembrin chalcone exhibited a reduction due to FR, while UVA promoted it. In FR treatment, the up-regulated metabolite included 2’,3,4,4’,6’-pentahydroxychalcone. Conversely, the down-regulated metabolites encompassed butein, phlorizin, and naringenin. Pinobanksin 3-acetate was down-regulated in response to UVA exposure (Figure 13). The analysis indicated that different LED light qualities induced complicated alterations in gene and metabolite expression within the phenylpropanoid and flavonoid biosynthesis pathways of mint.

### 2.7. Validation of RNA-Seq Based DEGs Results by qRT-PCR

The expression levels of three genes were analyzed using qRT-PCR to verify the reliability of the RNA-Seq data. FR improved plant growth by increasing the expression levels of *AUX/IAA* and *POD*. Mint significantly increased the expression level of *DELLA*, facilitating adaptation to UVA irradiation (Figure 14A). Figure 14B showed the Log_2_ Fold Change of RNA-seq and qRT-PCR analysis of *AUX/IAA*, *DELLA*, and *POD*. This indicated that the up-regulation trend of *AUX/IAA*, *DELLA*, and *POD* was consistent between RNA-seq and qRT-PCR. The validation results for qRT-PCR indicated that the RNA-Seq data were reliable.

### 2.8. Correlation Analysis

The correlation analysis indicates a significant positive correlation among plant growth indicators. Indicators of plant growth demonstrate a positive correlation with P_n_, ΦPSII, ETR, F_v_’/F_m_’, F_v_/F_m_, PI, *AUXIAA*, and *POD*. Conversely, they are significantly negatively correlated with Chl content, C_i_, and *DELLA* (Figure 15). PCA was conducted to better understand the effects of different treatments on plant growth.

PCA demonstrated the distribution and variation trends of various physiological parameters under different treatments (CK, FR, and UVA). PCA was performed on 24 indicators, which can be mainly divided into four principal components (PCs). These components collectively explain 94.0% of the variance. PC1 contributes 59.7% of the variance, PC2 accounts for 26.0%, PC3 explains 5.1%, and PC4 contributes 3.2% of the variance, collectively illustrating the changes in plant growth states under different treatment conditions. In PC1, the variables with larger loadings are plant height, plant width, biomass, PI, *AUX/IAA*, *POD*, and Chl content (Figure 16). In PC2, G_s_, T_r_, and WUE exhibit larger loadings (Figure 16). It can be seen that FR light increases F_v_’/F_m_’ and the expression of *AUX/IAA* and *POD*, which may enhance light trapping capacity in mint, subsequently promoting plant growth and biomass.

## 3. Discussion

### 3.1. Effects of Different Light Qualities on the Growth of Mint

Light is an indispensable environmental factor that significantly influence plant growth and development. Different light qualities, especially its spectral composition, greatly impact the morphological and physiological responses of plants. The study revealed a significant increase in plant height, plant width, and biomass of mint with the addition of FR light to RB light (Figure 2 and Figure 3). A recent study indicated that shade-avoidance syndrome (SAS) was triggered by FR light, leading to enhanced shoot elongation [24]. This response enables plants to capture more light energy for photosynthesis and to accumulate greater biomass. Consistently with our results, the addition of FR radiation increased the plant height and fresh weight of cucumber seedlings, as well as the shoot biomass of leaf lettuce [25]. FR radiation facilitated the accumulation of GA and IAA in stems, leading to stem elongation [26]. A prior study indicated that increased UVB radiation resulted in an average reduction of 15.5% in plant height, 16.9% in the dry weight of individual stems, and 43.7% in yield per plant of three soybean cultivars [27]. Our results consistently indicated that UVA light slightly inhibited mint growth, as evidenced by measurements of the biomass of mint (Figure 3). This effect may be attributed to the low PPFD and PPF of UVA in the experiment. The UVA treatment exhibited minimal effects on mint; therefore, the discussion will concentrate on the response mechanisms of mint plants to FR light treatment.

### 3.2. Effects of Different Light Qualities on the Chlorophyll and Carotenoid Contents of Mint

Plant pigments play a crucial role in photosynthesis and photoprotection, notably influencing plant growth and adaptation to environmental changes. Exposure to FR light resulted in a significant decrease in the concentrations of Chl and Car in lettuce leaves [28]. Our results indicated a significant decrease in Chl *a*, Chl *b*, and Car contents under FR treatment when compared to CK (Table 1). Leister’s research observed the potential for enhancing leaf light utilization through the reduction of Chl content, increase of photosynthetic protein levels, and the extension of the photosynthetic mechanism [29].

### 3.3. Effects of Different Light Qualities on the Photosynthesis of Mint

The vigor of plant photosynthetic capacity hinges on the P_n_ within its leaves, wherein light quality plays a crucial role in shaping this rate. Despite being outside the visible light spectrum, FR light is crucial for photosynthesis [30]. The supplementary FR light enhanced ΦPSII and P_n_ while decreasing the non-photochemical quenching of the fluorescence (NPQ) of lettuce (*Lactuca sativa*) [31]. The increase in P_n_ of mint under FR treatment contributed to the enhancement of its biomass, aligning with the findings of Zhou et al. [32]. This phenomenon may result from FR light enhancing plant height to optimize light energy capture, thereby improving photosynthetic capacity through the modulation of light energy direction and photosynthetic electron transport [33]. UVA treatment, however, inhibited P_n_, G_s_, and T_r_ (Figure 5). Other studies have consistently demonstrated that UVB radiation adversely affects the photosynthesis of almond seedlings, impacting CO_2_ assimilation, G_s_, and T_r_ [34]. The study conducted by Mackerness et al. indicated that chloroplasts were sacrificed to safeguard the remainder of the cell during UVB stress [35]. UV radiation may influence photosynthetic pigments by hindering their synthesis or the enzymatic activities within the chlorophyll biosynthetic pathway, thereby reducing the P_n_.

The photosynthetic process consists of closely interconnected steps whereby changes in the PSII center trigger alterations in fluorescence and subsequent photosynthesis. This study revealed that FR light significantly enhanced the efficiency of PSII electron transport in mint leaves, indicated by the increased F_v_’/F_m_’, which subsequently improved light use efficiency and photosynthetic capacity. Under UVA treatment, the ФPSII, ETR, and qP exhibited a reduction (Figure 6). Hazrati et al. found a significant decrease in F_v_/F_m_ and the potential activity of PSII (F_v_/F_o_) can be considered as indicators of photoinhibition [36]. The addition of FR radiation to lettuce increased F_v_/F_m_ and PI [37]. Our study demonstrated that F_v_/F_m_ and PI of mint increased under FR treatment, suggesting that FR efficiently improved the light energy conversion efficiency of PSII in mint leaves and improved plant health status (Figure 6).

### 3.4. Effects of Different Light Qualities on the Transcriptome and Metabolome of Mint

Different light qualities are crucial for the growth and physiological metabolism of plants. They serve as an energy source for photosynthesis and also alter gene expression levels and metabolite profiles in plants [38]. Prior research indicated that RB and UVB radiation markedly elevate the levels of flavonoids and anthocyanins, as well as the expression of *the basic leucine zipper* (*bZIP*) gene in strawberry fruit [39]. The expression of genes associated with the phenylpropanoid biosynthesis pathway was induced by light [40]. Our research revealed a notable difference in the quantity of DEGs and DAMs. DEGs and DAMs were primarily associated with plant hormone, phenylpropanoid, and flavonoid pathways. The results indicated that an appropriate increase in FR or UVA radiation might alter transcript reprogramming and redirect metabolic flux in mint, subsequently regulating its growth and secondary metabolism (Figure 8 and Figure 10).

#### 3.4.1. The Effects of Different Light Qualities on the Plant Hormone Signal Transduction Pathway in Mint

Light is closely related to hormone regulation in plant growth and development [41], and our results indicated that different light treatments have different effects on mint hormone levels (Figure 11). Our research indicated that different light treatments influenced the expression levels of genes and metabolites in mint growth variously. FR was observed to enhance the expression of *AUX/IAA*, a gene associated with IAA signaling, whereas UVA was found to inhibit it (Figure 11). IAA transport was recognized as a critical factor for FR-induced elongation [42]. A low R/FR ratio can increase IAA levels in various plant tissues and organs across different developmental stages. IAA translocated from leaves to stems resulted in elongated internode distance and increased plant height as part of the shade-avoiding response in plants [43]. The GA pathway, associated with growth, was significantly activated by FR treatment. *GID1* exhibited a significant up-regulation in FR (Figure 11). Previous study indicated that GA plays a role in FR-regulated petiole elongation in arabidopsis [44] and contributes to improved light capture efficiency and enhanced biomass accumulation in tobacco plants [45]. The expression of *DELLA*, a key GA signaling repressor, was significantly up-regulated in response to UVA (Figure 11). Research indicated that *DELLA* inhibits growth while simultaneously promoting JA signal transduction through its interaction with *JAZ* [46,47]. Different light qualities significantly impact plant adaptability. A decrease in photoactive radiation and a reduction in the red/far-red light ratio can induce SAS in plants, thereby reducing their resistance to abiotic and biotic stresses [48]. FR light enhanced the accumulation of JA, a hormone associated with plant defense [49]. Under FR treatment, there was an increased accumulation of the stress-responsive hormone ABA, along with elevated expression of *PP2C* (Figure 11). The findings indicated that the addition of FR enhanced the adaptability of mint. Consistently with our findings, the addition of moderate FR light elevated ABA levels during the cold acclimation of barley [50]. We speculated that the accumulation of ABA and JA alleviated the negative effects of SAS on mint with FR. UVA suppresses growth-promoting hormones while inducing defense-related hormones, facilitating mint’s adaptation to UV light exposure.

#### 3.4.2. The Effects of Different Light Qualities on the Phenylpropanoid Biosynthesis Pathway in Mint

The phenylpropanoid biosynthesis pathway is an important route linking primary and secondary metabolism in plants; phenylalanine serves as a substrate for many secondary metabolites, including phenylpropanoids, flavonoids, anthocyanins, lignin, and more. Phenylpropanoids are also the main sources of plant color and aroma, they help the plant to attract pollinators or deter enemies [51]. Phenylpropanoid biosynthesis begins with the formation of the aromatic amino acid phenylalanine. *PAL* catalyzes the conversion of phenylalanine to cinnamic acid. Subsequently, *Cinnamate 4-hydroxylase* (*C4H*) and *4-coumarate--CoA ligase* (*4CL*) catalyze the conversion of cinnamic acid to p-coumaroyl-CoA [52]. P-coumaroyl-CoA serves as a crucial branching point in phenylpropanoid biosynthesis, allowing entry into two major downstream pathways: lignin monomer and flavonoid biosynthesis [53]. Increased activities of *PAL* and other enzymes involved in the phenylpropanoid synthesis pathway, such as *POD* and polyphenol oxidase (*PPO*), may be responsible for the accumulation of phenolic and flavonoid compounds in leaves of cucumber seedlings [54]. The study indicated that FR enhanced the expression of *POD* within the phenylpropanoid biosynthesis pathway and facilitated the accumulation of sinapyl alcohol (Figure 12). Zhang et al. found that higher sinapyl alcohol contents lead to thicker cell walls in leaf tissues, underscoring the critical role of lignin in reinforcing the structural integrity of plant cells against stressors [55]. S-lignin has been identified as polymerized from sinapyl alcohol [56]. *PAL* is a key enzyme of the phenolic biosynthesis pathway [57]. A prior investigation indicated that phenolic metabolism triggered by FR treatment in silver birch seedlings could stem from resource allocation aimed at promoting rapid elongation growth [58]. Falcioni et al. found that increased GA and light levels donate to increased cell wall thickness and enhanced lignin deposition in xylem fibers [59]. Li’s research on arabidopsis indicated that red and FR light promote lignin biosynthesis, while exogenous GA also affects lignin synthesis [60]. It can be inferred that FR increased the elongation of mint hypocotyls via *AUXIAA* expression. GA concurrently stimulated *POD* enzyme activities, promoting the synthesis of sinapyl alcohol for S-lignin production. This process offered structural support to the cell wall, allowing for vertical growth and the transport of water and nutrients throughout its lifecycle [61]. FR promoted the accumulation of phenylalanine, sinapyl alcohol, methylchavicol, and anethole in the phenylpropanoid biosynthesis pathway (Figure 12). The research indicated that methylchavicol and anethole are significant components in Korean mint [62].

#### 3.4.3. The Effects of Different Light Qualities on the Flavonoid Biosynthesis Pathway in Mint

Flavonoids form a branch of phenylpropanoids. The first dedicated enzyme in the flavonoid pathway is *CHS*, which catalyzes the stepwise condensation of three molecules of malonyl-CoA to one molecule of 4-coumaroyl-CoA to yield naringenin chalcone. Subsequently, *chalcone isomerase* (*CHI*) catalyzes the isomerization of the chalcone to naringenin. The following reaction, catalyzed by *flavanone 3-hydroxylase* (*F3H*), yields dihydrokaempferol, which is further converted to dihydroquercetin and dihydromyricetin by the actions of *flavonoid 3′-hydroxylase* (*F3′H*) and *F3′5′H*, respectively. Consecutive reactions catalyzed by *DFR* and *anthocyanidin synthase* (*ANS*) convert the dihydroflavonols to colored anthocyanidin aglycones, leading to a large repertoire of species-specific anthocyanins [63]. Flavonoids are important secondary metabolites in mint. As a source of various antioxidants, flavonoids have important economic value in food, spices, medicine, and other fields. During our experiments, an increase in the expression of flavonoid biosynthesis genes was also noted. FR up-regulated the expression of *CHS* and promoted the accumulation of luteolin, a natural antioxidant, and leucocyanidin in the flavonoid biosynthesis pathway. The signaling response induced by UVA initiated *HCT*, *DFR*, and *C12RT1* in the flavonoid biosynthesis pathway (Figure 13). The mechanisms underlying the synthesis of secondary metabolites in mint, influenced by different light qualities, remain unclear and warrant further investigation.

## 4. Materials and Methods

### 4.1. Plant Material and Treatments

The experiment was conducted in a plant factory at the Chongming base of the National Engineering Research Center of Protected Agriculture from August to October 2023. Mint seeds, purchased from Beijing Fengming Yashi Technology Development Co., Ltd., were selected and soaked in 55 °C warm water for 5 h to facilitate germination. After germination, the seeds were sown in 128-cell trays with coconut coir as the cultivation substrate.

The 20-day-old well-grown mint plants, approximately 2.7 cm in height, were subsequently treated with different light qualities on the seedling rack (Figure 17A). The seedbed was positioned 33 cm from the lamp, with an initial PPFD value of 221 µmol m^−2^ s^−1^, at which time the lamp was 25 cm away from the plants (Table 3). The lighting passport (ALP-01, Asensetek, Taipei, Taiwan, China) was used to measure the PPFD parameters. The PPF values of different treatments are shown in Table 4, spectral graph is shown in Figure 17B. The experiment was designed with eight plants for each treatment with three biological replications. Three treatments were designed in this experiment: (1) 7R3B (CK): seedlings grown under LED with a 7:3 ratio of red to blue light; (2) 7R3B+FR (FR): seedlings grown under LED with a 7:3 ratio of red to blue light and an increase in far-red light; (3) 7R3B+UVA (UVA): seedlings grown under LED with a 7:3 ratio of red to blue light and an increase in ultraviolet light A. The temperature for all the experiments was set to 25 °C as the day (13 h) and 23 °C as night (11 h). The relative humidity of all the experiments was 75 ± 10%. Irrigation was carried out using a nutrient solution with a pH of around 5.5 and an electrical conductivity (EC) of around 2.0 ms cm^−1^. The largest leaves of each treatment were harvested and immediately frozen in liquid nitrogen and stored at −80 °C for further analysis after 34 days of the different light quality treatments.

### 4.2. Growth Parameters Measurement

Five mint seedlings were randomly selected in each treatment for measurement of the morphological index. Plant height (the distance from the base to the growing point above the ground), plant width (the maximum width that the aboveground parts of plants can form), stem diameter, leaf length and leaf width (the above and middle fully developed leaves) were measured using a ruler. The plants were carefully removed from the substrate, the roots were washed, excess water was absorbed with paper, and after determining the fresh weight of the shoots and roots parts, they were placed in a drying oven at 120 °C for 2 h for blanching, followed by drying at 80 °C for 3 days. The dry weight of shoots and roots were measured using an electronic scale with an accuracy of 0.0001.

### 4.3. Measurement of Chlorophyll and Carotenoid Contents

The fresh leaf tissues (0.1 g) were taken from the leaves and soaked in 10 mL of 95% ethanol. After the leaves faded to white, the absorbance of the supernatant was measured at 663, 645 and 470 nm using a UV–visible spectrophotometer (Ultraviolet-2700; Shimadzu, Tokyo, Japan), respectively. The Chl *a*, Chl *b*, and Car contents were calculated following Lichtenthaler & Wellburn [64]:$$C_{a}=13.95\;A_{665}-6.88\;A_{649}\;C_{b}=24.96\;A_{649}-7.32\;A_{665}\;C_{x\;c}=\frac{1000\;A_{470}-2.05\;C_{a}-114.8\;C_{b}}{245}$$

In the formula, *C_a_* and *C_b_* represent the concentrations of Chl *a* and Chl *b*, respectively; *C_x c_* is the total concentration of Car; *A*_665_, *A*_649_, and *A*_470_ represent the absorbance of chloroplast pigment extract at wavelengths of 665 nm, 649 nm, and 470 nm, respectively.

### 4.4. Leaf Gas Exchange and Chlorophyll Fluorescence Analysis, Fitting of the Light Response Curve

Leaf gas exchange parameters were determined according to the procedure reported previously [16]. The portable photosynthesis system (CIRAS-3, PP Systems, Amesbury, MA, USA) equipped with a leaf chamber fluorometer (PLC3 Universal Leaf Cuvette, 18 × 25 mm window, CFM-3) was used to measure the gas exchange parameters and chlorophyll fluorescence of the fully expanded functional leaves of mint, including net photosynthetic rate (P_n_), intercellular CO_2_ concentration (C_i_), stomatal conductance (G_s_), transpiration rate (T_r_), water use efficiency (WUE), actual photochemical efficiency of PSII (ΦPSII), effective quantum yield of PSII photochemistry (F_v_’/F_m_’), electron transport rate (ETR), and photochemical quenching coefficient (qP). The irradiance level was set at 1000 µmol m^−2^ s^−1^, while temperature, humidity, and CO_2_ concentration were maintained at natural conditions in the plant factory. The leaves were acclimated to the irradiance level for approximately 1–2 min prior to the measurements. As described by Jiang et al. [65], chlorophyll fluorescence measurements were performed using the same leaves as the gas exchange measurements. Maximal quantum yield of PSII (F_v_/F_m_) and performance index (PI) were measured after dark adaptation using a Pocket PEA chlorophyll fluorescence meter produced by Hansatech (King’s Lynn, Norfolk, UK). Gas exchange and chlorophyll fluorescence parameters were measured on the same leaf blade with five replicates of each treatment. Results were presented as the mean ± standard deviation. Light response curve was measured using the portable photosynthesis system (CIRAS-3, PP Systems, Amesbury, MA, USA) on the fully expanded functional leaves of mint. Irradiance levels were set at 0, 10, 20, 50, 80, 100, 120, 150, 200, 400, 600, 800, 1000, 1200, 1500, and 1800 μmol m^−2^ s^−1^, leaves were allowed to acclimate to each irradiance level for about 2 min before measurement, and irradiance was increased gradually from 0 to 1800 μmol m^−2^ s^−1^. The fitting of the light response curve was performed using the photosynthesis calculation software developed by Ye [66] at https://photosynthetic.sinaapp.com, accessed on 11 March 2024.

### 4.5. Transcriptome cDNA Library Preparation and Sequencing and Real-Time Quantitative RT-PCR

#### 4.5.1. Transcriptome cDNA Library Preparation and Sequencing

Total RNA samples were processed for quality assessment of their purity, concentration and integrity. Measurements of RNA purity and concentration were performed using the NanoDrop 2000 spectrophotometer (Thermo Scientific, Waltham, MA, USA), and measurement of RNA integrity was performed using the Agient Bioanalyzer 2100 (Agilent Technologies, Santa Clara, CA, USA). Qualified RNA samples were processed for library preparation. Library concentration was initially measured by Qubit 3.0 (Thermo Fisher Scientific, Waltham, MA, USA). The concentration is expected to be larger than 1 ng μL ^−1^. Insert size distribution of the libraries was examined by Qsep400 (Bioptic, Changzhou, Jiangsu, China). A more precise measurement on valid library concentration was measured by Q-PCR on libraries passed insert check. Valid library concentration is expected to be higher than 2 nM. The qualified library was sequenced by the high-throughput sequencing platform with Illumina NovaSeq6000, PE150 mode (Illumina, San Diego, CA, USA). Raw reads were further processed using the bioinformatics analysis platform BMKCloud (www.biocloud.net).

#### 4.5.2. Real-Time Quantitative RT-PCR

The expression levels of *AUX/IAA* (TRINITY_DN23381_c0_g1), *DELLA* (TRINITY_DN9042_c0_g2), and *POD* (TRINITY_DN11178_c0_g1) genes were analyzed using qRT-PCR under CK, FR, and UVA treatments, to verify the reliability of the RNA-Seq data.

RNA extraction: Total RNA was extracted from mint leaf using AM1912 RNAqueous^®^ Kit (Thermo Fisher Scientific, Waltham, MA, USA) according to the manufacturer’s specifications. The yield of RNA was determined using a NanoDrop 2000 spectrophotometer (Thermo Scientific, Waltham, MA, USA), and the integrity was evaluated using agarose gel electrophoresis stained with ethidium bromide.

Real-time quantitative RT-PCR: Quantification was performed with a two-step reaction process: reverse transcription (RT) and PCR. Each RT reaction consisted of 0.5 μg RNA, 2 μL of 5 × TransScript All-in-one SuperMix for qPCR and 0.5 μL of gDNA Remover, in a total volume of 10 μL. Reactions were performed in a GeneAmp^®^ PCR System 9700 (Applied Biosystems, Carlsbad, CA, USA) for 15 min at 42 °C, 5 s at 85 °C. The 10 μL RT reaction mix was then diluted × 10 in nuclease-free water and held at −20 °C. Real-time PCR was performed using LightCycler^®^ 480 II Real-time PCR Instrument (Roche, Switzerland) with 10 μL PCR reaction mixture that included 1 μL of cDNA, 5 μL of 2 × PerfectStartTM Green qPCR SuperMix, 0.2 μL of forward primer, 0.2 μL of reverse primer and 3.6 μL of nuclease-free water. Reactions were incubated in a 384-well optical plate (Roche, Switzerland) at 94 °C for 30 s, followed by 45 cycles of 94 °C for 5 s, 60 °C for 30 s. Each sample was run in triplicate for analysis. At the end of the PCR cycles, melting curve analysis was performed to validate the specific generation of the expected PCR product. The primer sequences were designed in the laboratory and synthesized by TsingKe Biotech based on the mRNA sequences obtained from the NCBI database as Table 5. The expression levels of mRNAs were normalized to GAPDH (JN587699.1) and were calculated using the 2^−ΔΔCt^ method [67].

### 4.6. Metabolite Extraction, Detection, and Qualitative and Quantitative Analysis

Sample extraction: (1). Take out the sample stored at −80 °C, thaw it at room temperature, weigh 50 mg of the sample and add 1000 μL of extraction solution containing internal standards. Methanol (CAS: 67-56-1; pureness: LC-MS; Merck, Darmstadt, Germany): acetonitrile (CAS: 75-05-8; pureness: LC-MS; Merck, Darmstadt, Germany): water = 2:2:1, internal standard is 2-Chloro-L-phenylalanine (CAS: 64-18-6; pureness: LC-MS; Aladdin, Shanghai, China), internal standard concentration is 20 mg L^−1^. Vortex mix for 30 s. (2). Add steel beads and process in a grinder at 45 Hz for 10 min, followed by sonication for 10 min in an ice-water bath. (3). Stand at minus 20 °C for 1 h. (4). Centrifuge the sample at 4 °C and 12,000 rpm for 15 min. (5). Carefully remove 500 μL supernatant into the EP tube. (6). Dry the extract in a vacuum concentrator. (7). Add 160 μL of extraction solution (acetonitrile:water = 1:1) to the dried metabolites for reconstitution (in order to ensure that the QC sample volume meets the injection requirements after configuration). (8). Vortex for 30 s and sonicate in an ice-water bath for 10 min. (9). Centrifuge the sample at 4 °C and 12,000 rpm for 15 min. (10). Carefully take out 120 μL supernatant in 2 mL injection bottle and take 10 μL of each sample and mix it into QC sample for machine testing.

The LC-MS system for metabolomics analysis is composed of Waters Acquity I-Class PLUS ultra-high performance liquid tandem Waters Xevo G2-XS QTof (Waters, Milford, MA, USA) high resolution mass spectrometer. The column used is purchased from Waters Acquity UPLC HSS T3 column (1.8 μm 2.1 × 100 mm) (Waters, Milford, MA, USA). Positive ion mode: mobile phase A: 0.1% formic acid aqueous solution; mobile phase B: 0.1% formic acid acetonitrile. Negative ion mode: mobile phase A: 0.1% formic acid aqueous solution; mobile phase B: 0.1% formic acid acetonitrile. The elution gradients are shown in Table 6. Column temperature 50 °C, injection volume 1 μL. (Formic acid CAS: 64-18-6; pureness: LC-MS; TCI Shanghai, Shanghai, China)

Waters Xevo G2-XS QTOF high resolution mass spectrometer can collect primary and secondary mass spectrometry data in MSe mode under the control of the acquisition software (MassLynx V4.2, Waters, Milford, MA, USA). In each data acquisition cycle, dual-channel data acquisition can be performed on both low collision energy and high collision energy at the same time. The low collision energy is 2 V, the high collision energy range is 10~40 V, and the scanning frequency is 0.2 s for a mass spectrum. The parameters of the ESI ion source are as follows: Capillary voltage: 2000 V (positive ion mode) or −1500 V (negative ion mode); cone voltage: 30 V; ion source temperature: 150 °C; desolvent gas temperature 500 °C; backflush gas flow rate: 50 L h^−1^; Desolventizing gas flow rate: 800 L h^−1^; Collection range of mass charge ratio (*m*/*z*): 50–1200. (No UV absorption, as there is no corresponding analyzer in series, MS resolution is 25,000–35,000).

The raw data collected using MassLynx V4.2 (Waters, Milford, MA, USA) is processed by Progenesis QI V2.5 for peak extraction, peak alignment and other data processing operations, identification based on the Progenesis QI V2.5 online METLIN database and Biomark’s self-built library, and at the same time, theoretical fragment identification and mass deviation are all within 100 ppm.

After normalizing the original peak area information with the total peak area, the follow-up analysis was performed. Principal component analysis and Spearman correlation analysis were used to judge the repeatability of the samples within the group and the quality control samples. The identified compounds are searched for classification and pathway information in KEGG, HMDB and lipid maps databases. According to the grouping information, calculation and comparison of the difference multiples, a T test was used to calculate the difference significance *p* value of each compound. The R language package ropls was used to perform OPLS-DA modeling, and a 200 times permutation test was performed to verify the reliability of the model. The VIP value of the model was calculated using multiple cross-validation. The method of combining the difference multiple, the *p* value and the VIP value of the OPLS-DA model was adopted to screen the differential metabolites. The screening criteria are FC > 1, *p* value < 0.05 and VIP > 1. The difference metabolites of KEGG pathway enrichment significance were calculated using hypergeometric distribution test.

### 4.7. Statistical Analysis

Data organization was conducted using Excel 2021 (Microsoft, Redmond, WA, USA). Analysis of variance (ANOVA) was performed using SPSS 26.0 (IBM Corp, Armonk, NY, USA), with each value presented as mean ± standard deviation, based on three replicates. Graphs were generated using Origin 2024 (OriginLab Corporation, Northampton, MA, USA), Adobe Photoshop CS6 (Adobe Systems, San Jose, CA, USA) and Figdraw (http://www.figdraw.com).

## 5. Conclusions

Our study demonstrated that FR-treated mint exhibited higher plant height, plant width, biomass, F_v_’/F_m_’, F_v_/F_m_, and PI. The process of mint responding to FR irradiation involved the synergistic interaction of multiple pathways. Through the analysis of three key pathways, it was found that the enrichment of DEGs and DAMs in plant hormone signal transduction, phenylpropanoid biosynthesis, and flavonoid biosynthesis pathways plays an important role in promoting mint growth, development, and secondary metabolism. The findings of this study improved our understanding of the mechanisms by which FR and UVA light affect the growth and development of mint and provide effective suggestions for the combination of light quality in mint cultivation practices, which can help optimize its growth and increase yield (Figure 18).

## Figures and Tables

**Figure 1 plants-13-03495-f001:**
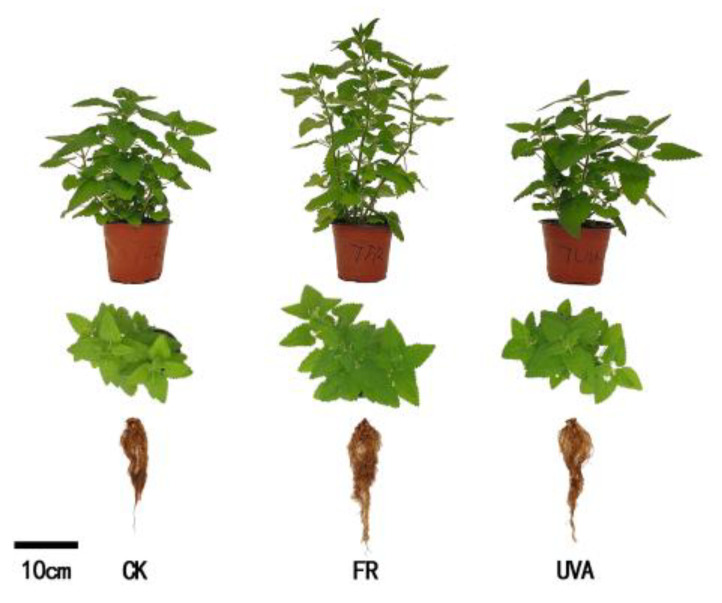
Effects of different light qualities on mint growth morphology. CK: 70% red light and 30% blue light (7R3B), FR: 7R3B + far-red light, UVA: 7R3B + ultraviolet light A.

**Figure 2 plants-13-03495-f002:**
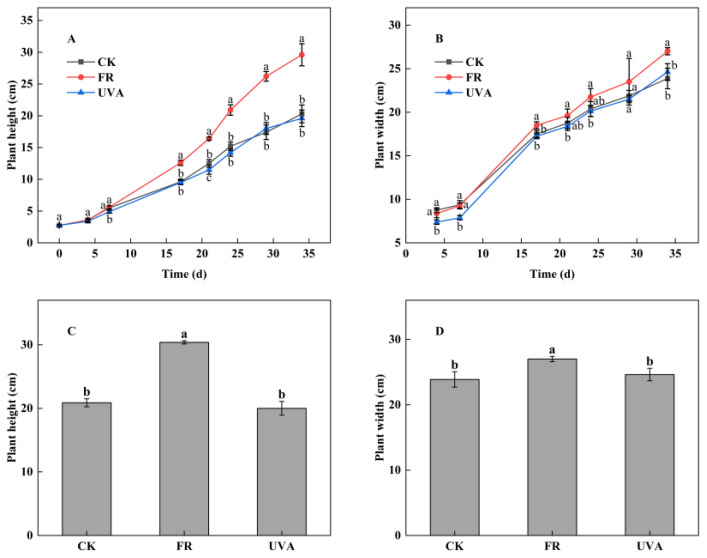
Effects of different light qualities on mint growth. (**A**) Plant height. (**B**) Plant width. (**C**) Plant height (d34). (**D**) Plant width (d34). CK: 70% red light and 30% blue light (7R3B), FR: 7R3B + far-red light, UVA: 7R3B + ultraviolet light A. Treatments were replicated three times, and different letters indicated significant differences (*p* < 0.05).

**Figure 3 plants-13-03495-f003:**
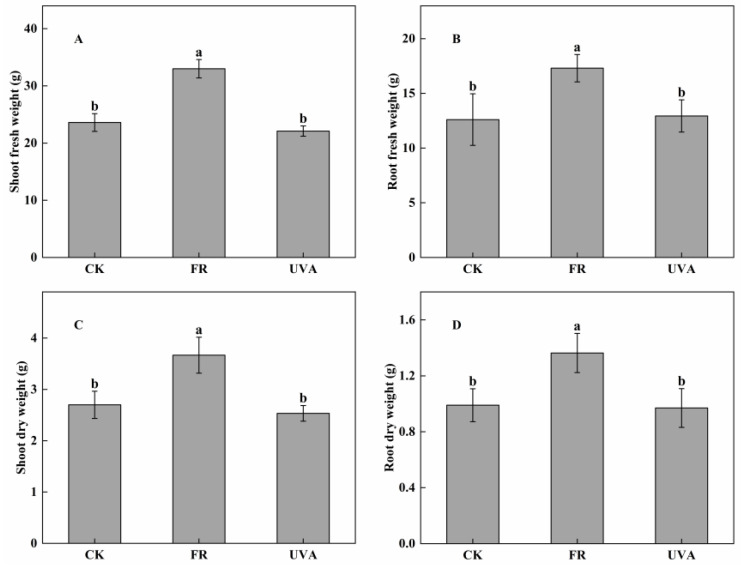
Effects of different light qualities on mint biomass. (**A**) Shoot fresh weight. (**B**) Root fresh weight. (**C**) Shoot dry weight. (**D**) Root dry weight. CK: 70% red light and 30% blue light (7R3B), FR: 7R3B + far-red light, UVA: 7R3B + ultraviolet light A. Treatments were replicated three times, and different letters indicated significant differences (*p* < 0.05).

**Figure 4 plants-13-03495-f004:**
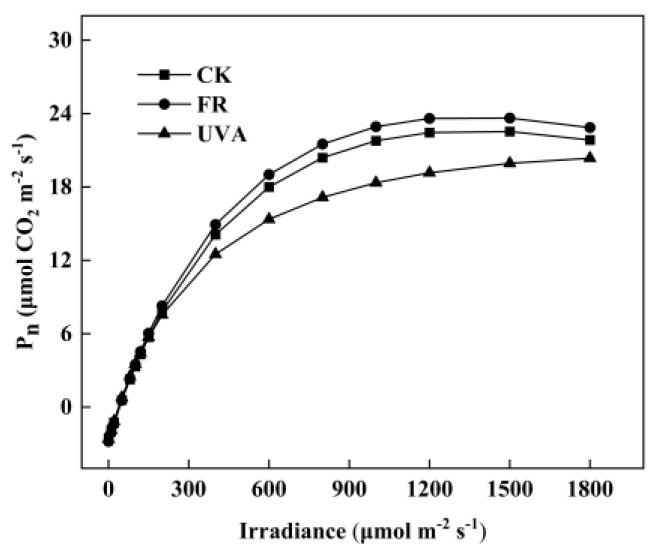
Effects of different light qualities on mint light response curves. CK: 70% red light and 30% blue light (7R3B), FR: 7R3B + far-red light, UVA: 7R3B + ultraviolet light A.

**Figure 5 plants-13-03495-f005:**
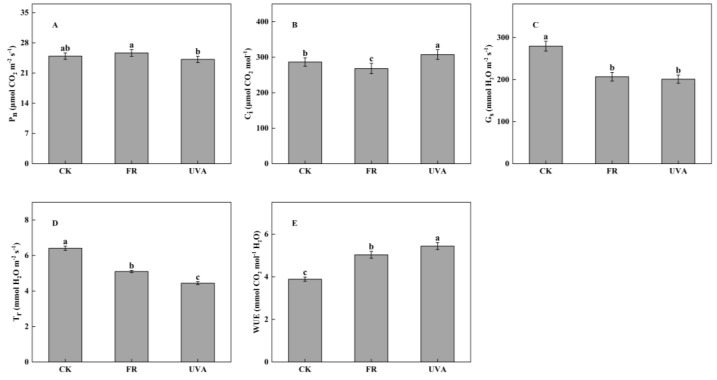
Effects of different light qualities on mint gas exchange parameters. (**A**) Net photosynthetic rate (P_n_). (**B**) Intercellular CO_2_ concentration (C_i_). (**C**) Stomatal conductance (G_s_). (**D**) Transpiration rate (T_r_). (**E**) Water use efficiency (WUE). CK: 70% red light and 30% blue light (7R3B), FR: 7R3B + far-red light, UVA: 7R3B + ultraviolet light A. Treatments were replicated three times, and different letters indicated significant differences (*p* < 0.05).

**Figure 6 plants-13-03495-f006:**
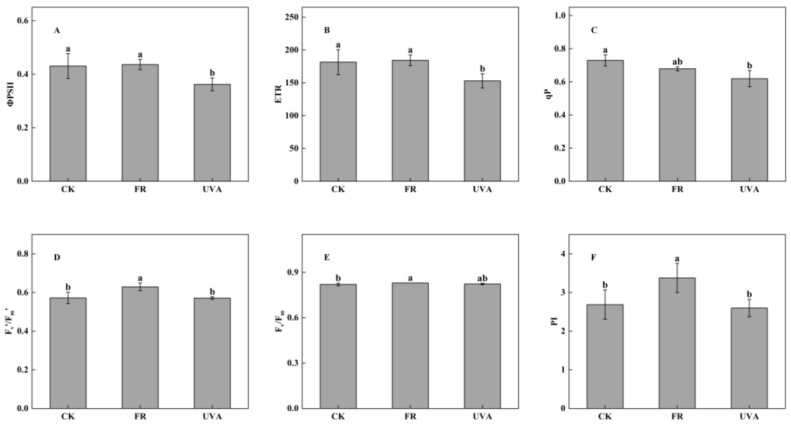
Effects of different light qualities on chlorophyll fluorescence parameters. (**A**) Actual photochemical efficiency of PSII (ΦPSII). (**B**) Electron transport rate (ETR). (**C**) Photochemical quenching coefficient (qP). (**D**) Effective quantum yield of PSII photochemistry (F_v_’/F_m_’). (**E**) Maximal quantum yield of PSII (F_v_/F_m_). (**F**) Performance index (PI). CK: 70% red light and 30% blue light (7R3B), FR: 7R3B + far-red light, UVA: 7R3B + ultraviolet light A. Treatments were replicated three times, and different letters indicated significant differences (*p* < 0.05).

**Figure 7 plants-13-03495-f007:**
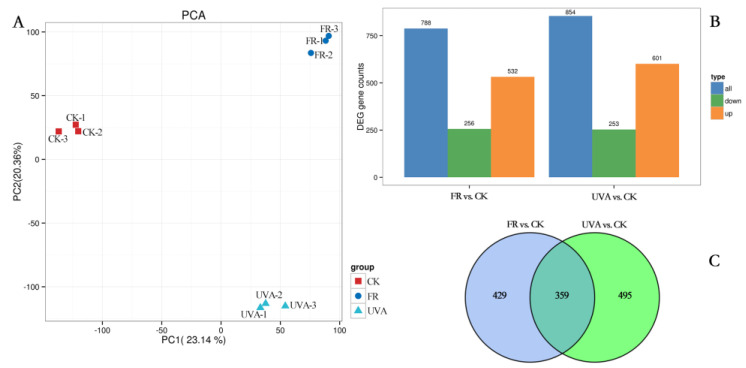
Effects of different light qualities on gene expression. (**A**) PCA of transcriptome samples under different light qualities. (**B**) Number of DEGs detected in FR vs. CK and UVA vs. CK. (**C**) Venn diagram of DEGs in FR vs. CK and UVA vs. CK. CK: 70% red light and 30% blue light (7R3B), FR: 7R3B + far-red light, UVA: 7R3B + ultraviolet light A. Treatments were replicated three times.

**Figure 8 plants-13-03495-f008:**
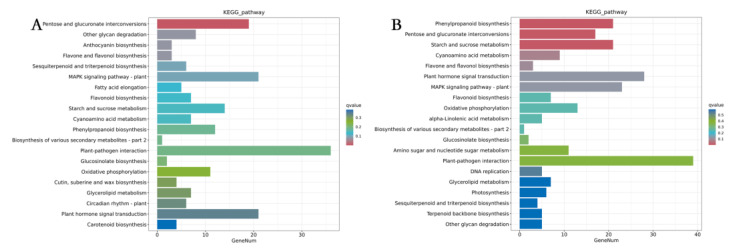
Effects of different light qualities on KEGG pathway enrichment of DEGs. (**A**) KEGG pathway enrichment of DEGs between FR and CK treatments. (**B**) KEGG pathway enrichment of DEGs between UVA and CK treatments. CK: 70% red light and 30% blue light (7R3B), FR: 7R3B + far-red light, UVA: 7R3B + ultraviolet light A. Treatments were replicated three times.

**Figure 9 plants-13-03495-f009:**
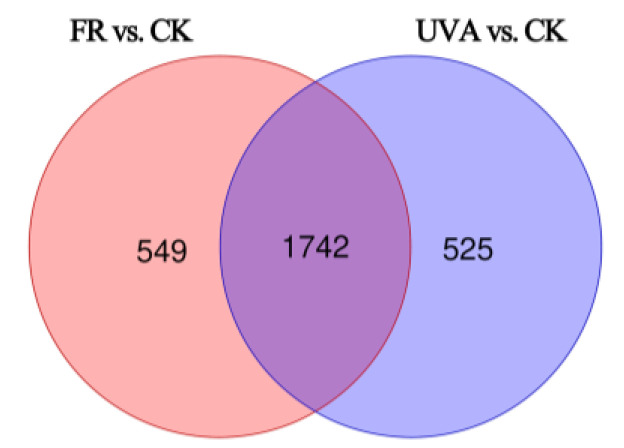
Venn diagram of DAMs in FR vs. CK and UVA vs. CK. CK: 70% red light and 30% blue light (7R3B), FR: 7R3B + far-red light, UVA: 7R3B + ultraviolet light A. Treatments were replicated three times.

**Figure 10 plants-13-03495-f010:**
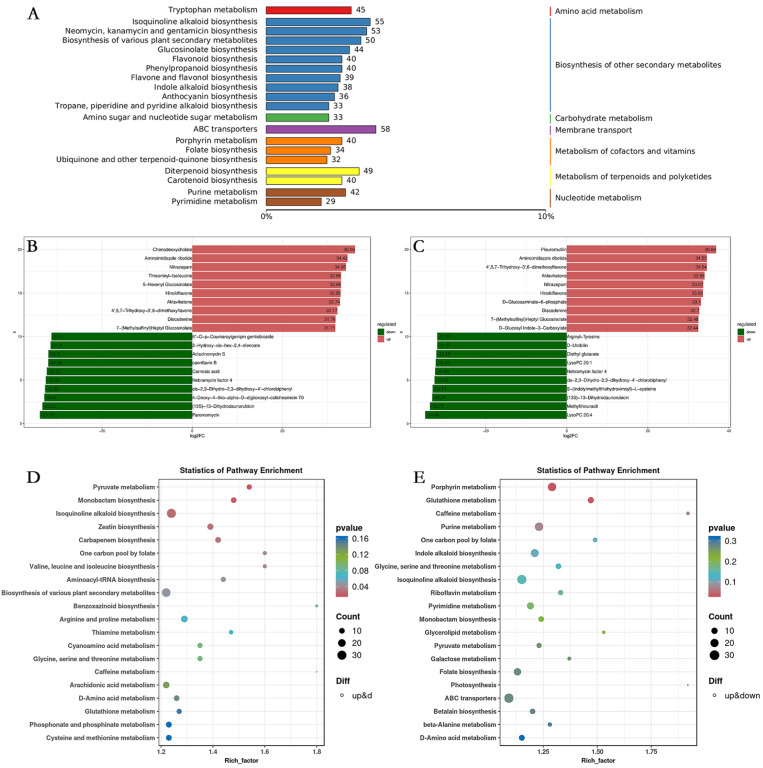
Effects of different light qualities on KEGG pathway enrichment of DAMs. (**A**) KEGG annotation of DAMs. (**B**) The top ten up and down-regulated DAMs of fold change between FR and CK treatments. (**C**) The top ten up and down-regulated DAMs of fold change between UVA and CK treatments. (**D**) KEGG pathway enrichment of DAMs between FR and CK treatments. (**E**) KEGG pathway enrichment of DAMs between UVA and CK treatments. CK: 7R3B, FR: 7R3B + far-red light, UVA: 70% red light and 30% blue light (7R3B) + ultraviolet light A. Treatments were replicated three times.

**Figure 11 plants-13-03495-f011:**
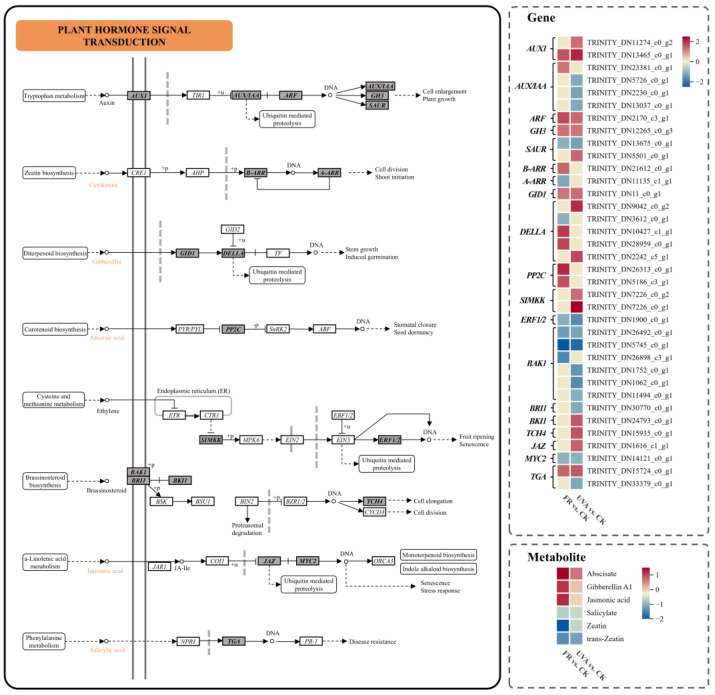
The DEGs and DAMs involved in plant hormone signal transduction pathway in response to different light qualities. The color in the rectangle represents the genes or metabolites that were regulated under different light qualities (red indicated up-regulation; yellow indicated non-significant; blue indicated down-regulation). CK: 70% red light and 30% blue light (7R3B), FR: 7R3B + far-red light, UVA: 7R3B + ultraviolet light A. Treatments were replicated three times.

**Figure 12 plants-13-03495-f012:**
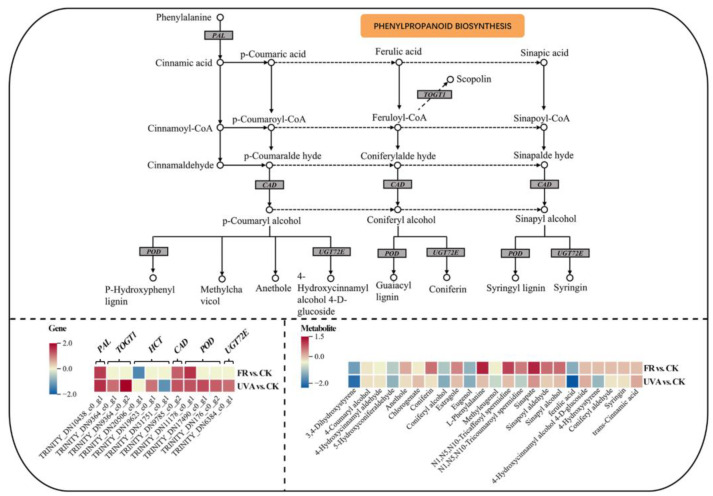
The DEGs and DAMs involved in phenylpropanoid biosynthesis pathway in response to different light qualities. The color in the rectangle represents the genes or metabolites that were regulated under different light qualities (red indicated up-regulation; yellow indicated non-significant; blue indicated down-regulation). CK: 70% red light and 30% blue light (7R3B), FR: 7R3B + far-red light, UVA: 7R3B + ultraviolet light A. Treatments were replicated three times.

**Figure 13 plants-13-03495-f013:**
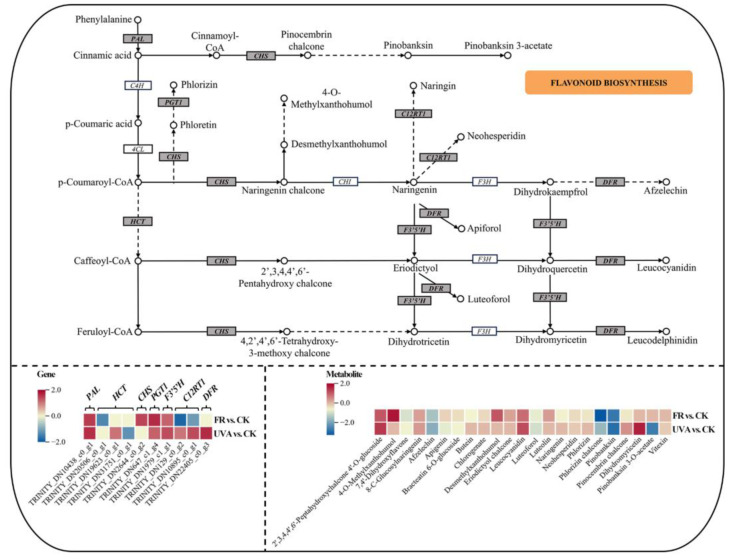
The DEGs and DAMs involved in flavonoid biosynthesis pathway in response to different light qualities. The color in the rectangle represents the genes or metabolites that were regulated under different light qualities (red indicated up-regulation; yellow indicated non-significant; blue indicated down-regulation). CK: 70% red light and 30% blue light (7R3B), FR: 7R3B + far-red light, UVA: 7R3B + ultraviolet light A. Treatments were replicated three times.

**Figure 14 plants-13-03495-f014:**
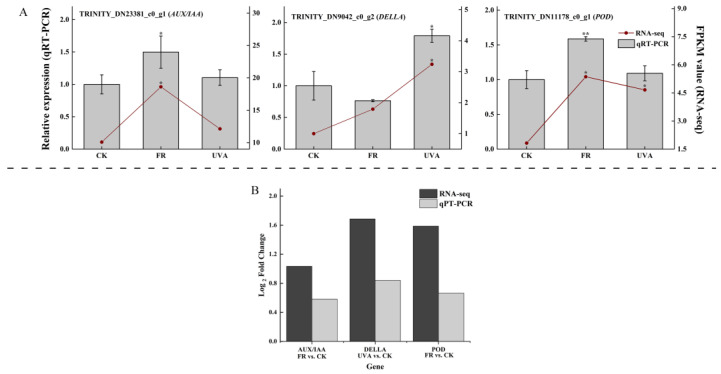
(**A**) qRT-PCR analysis of the gene expression patterns and FPKM expression level in mint seedlings under different light qualities. (**B**) Log_2_ Fold Change of RNA-seq and qRT-PCR analysis of *AUX/IAA*, *DELLA*, and *POD*. CK: 70% red light and 30% blue light (7R3B), FR: 7R3B + far-red light, UVA: 7R3B + ultraviolet light A. Treatments were replicated three times. “*” indicates a significant correlation at *p* ≤ 0.05 and “**” indicates a significant correlation at *p* ≤ 0.01.

**Figure 15 plants-13-03495-f015:**
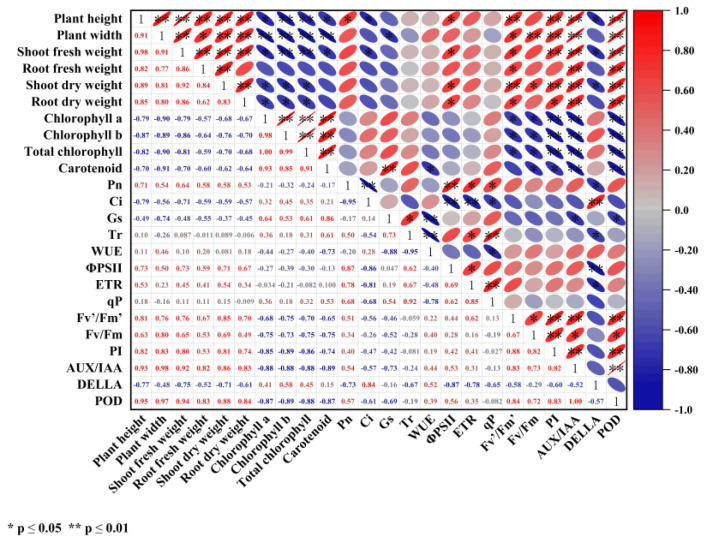
Correlation analysis between different results. The upper right ellipse represents the correlation between different parameters, and the lower left numbers represent the correlation coefficients, with red being a positive correlation and blue being a negative correlation. “*” indicates a significant correlation at *p* ≤ 0.05 and “**” indicates a significant correlation at *p* ≤ 0.01.

**Figure 16 plants-13-03495-f016:**
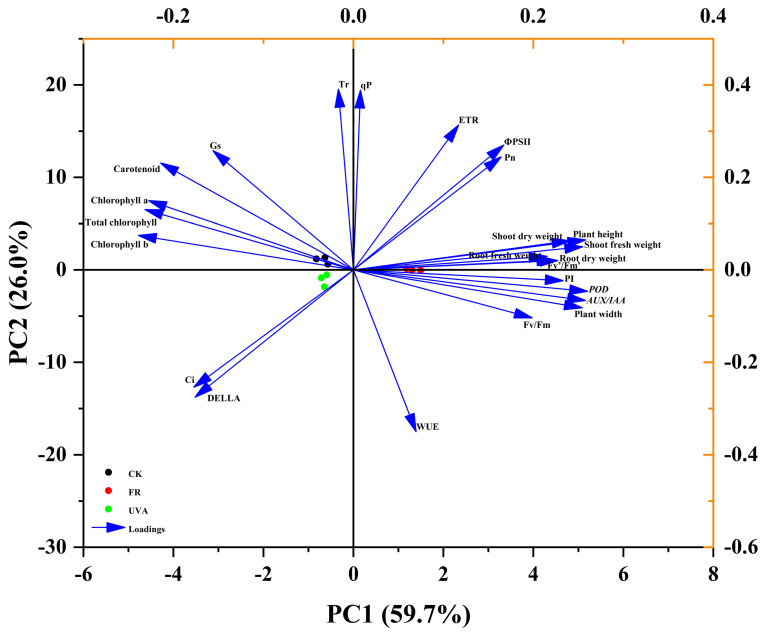
PCA of different results. Arrow direction and length indicate correlation and strength, respectively.

**Figure 17 plants-13-03495-f017:**
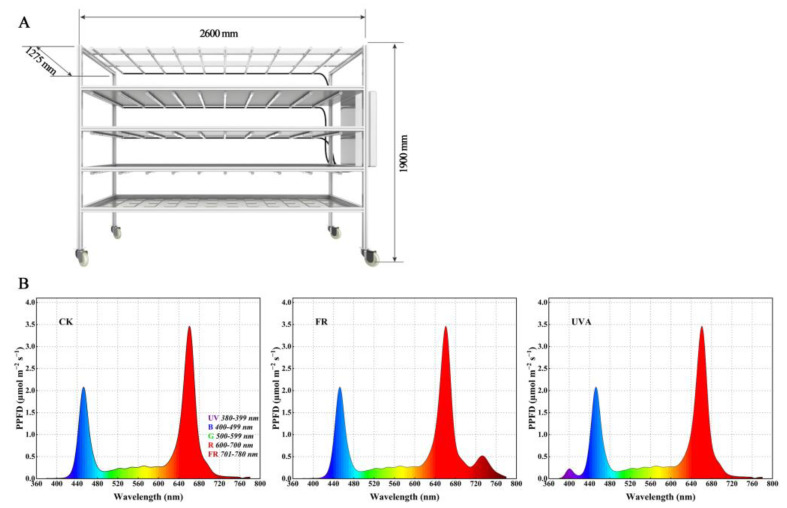
(**A**) Seedling rack for conducting experiments. (**B**) Spectral graphs of different treatments. CK: 70% red light and 30% blue light (7R3B), FR: 7R3B + far-red light, UVA: 7R3B + ultraviolet light A.

**Figure 18 plants-13-03495-f018:**
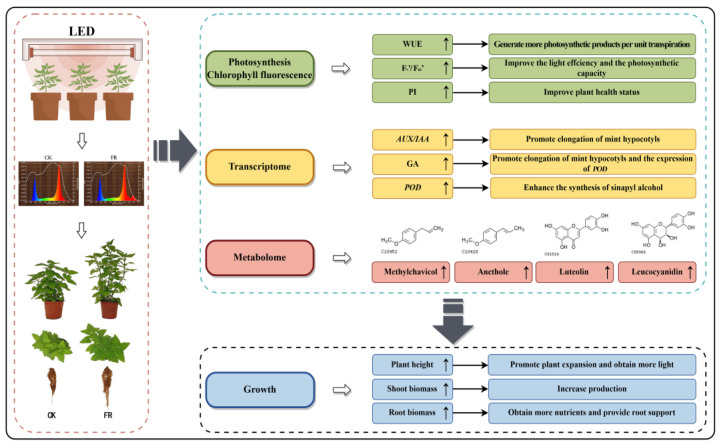
The frame diagram depicting the addition of FR to red and blue light on the growth, photosynthesis, transcriptome, and metabolome of mint. The arrow “↑” indicates that the indicator was up-regulated under FR. FR: 70% red light and 30% blue light (7R3B) + far-red light.

**Table 1 plants-13-03495-t001:** Effects of different light qualities on mint chlorophyll and carotenoid contents.

Treatment	Chlorophyll *a* (mg g^−1^)	Chlorophyll *b* (mg g^−1^)	Total Chlorophyll (mg g^−1^)	Carotenoid (mg g^−1^)
CK	1.61 ± 0.05 a	0.54 ± 0.01 a	2.15 ± 0.06 a	0.28 ± 0.01 a
FR	1.45 ± 0.01 b	0.49 ± 0.01 b	1.94 ± 0.02 b	0.22 ± 0.01 c
UVA	1.57 ± 0.06 a	0.54 ± 0.02 a	2.11 ± 0.08 a	0.25 ± 0.01 b

Note: CK: 70% red light and 30% blue light (7R3B), FR: 7R3B + far-red light, UVA: 7R3B + ultraviolet light A. Treatments were replicated three times, and different letters indicated significant differences (*p* < 0.05).

**Table 2 plants-13-03495-t002:** Statistical table of DAMs in mint leaves under different light qualities.

Group	Diff Number	Up Number	Down Number
FR vs. CK	2291	1150	1141
UVA vs. CK	2267	911	1356

Note: Diff Number: number of metabolites with significant differences. Up Number: up-regulation of the number of metabolites. Down Number: down-regulation of the number of metabolites. CK: 70% red light and 30% blue light (7R3B), FR: 7R3B + far-red light, UVA: 7R3B + ultraviolet light A. Treatments were replicated three times.

**Table 3 plants-13-03495-t003:** PPFD radiation parameters of different treated seedlings at the beginning of treatment.

Treatment	PPFD, µmol m^−2^ s^−1^ of Spectral Ranges (nm)					
	Total	UV	Blue	Green	Red	Far-Red
	(380–780)	(380–399)	(400–499)	(500–599)	(600–699)	(700–780)
	(µmol m^−2^ s^−1^)	(µmol m^−2^ s^−1^)	(µmol m^−2^ s^−1^)	(µmol m^−2^ s^−1^)	(µmol m^−2^ s^−1^)	(µmol m^−2^ s^−1^)
CK	221.6	0.13	67.9	28.4	125.4	5.6
FR	221.9	0.09	68.0	28.5	125.4	27.7
UVA	224.1	1.85	70.7	28.2	125.2	5.4

Note: Measure at 28 cm away from the LED tube. CK: 70% red light and 30% blue light (7R3B), FR: 7R3B + far-red light, UVA: 7R3B + ultraviolet light A.

**Table 4 plants-13-03495-t004:** PPF parameters of different treatments.

Treatment	PPF, µmol s^−1^ of Spectral Ranges (nm)					
	Total	UV	Blue	Green	Red	Far-Red
	(380–780)	(380–399)	(400–499)	(500–599)	(600–699)	(700–780)
	(µmol s^−1^)	(µmol s^−1^)	(µmol s^−1^)	(µmol s^−1^)	(µmol s^−1^)	(µmol s^−1^)
CK	848.016	0.0480	224.760	94.068	529.080	0.060
FR	953.496	0.024	224.040	93.480	530.160	105.792
UVA	873.192	16.908	231.600	93.336	531.240	0.108

Note: The red-blue ratio is the energy ratio of red and blue light in this combined spectrum (within the range of 380–780 nm); FR increases infrared light by 13% on the basis of the total PPF while keeping the red/blue ratio constant; UVA increases ultraviolet light by 2% on the basis of the total PPF while keeping the red blue ratio constant. CK: 70% red light and 30% blue light (7R3B), FR: 7R3B + far-red light, UVA: 7R3B + ultraviolet light A.

**Table 5 plants-13-03495-t005:** Details on designed primer sets.

Gene Symbol	Forward Primer (5’ to 3’)	Reverse Primer (5’ to 3’)	Product Length (bp)	Tm (°C)
GAPDH (JN587699.1)	TGATGTCTCTGTGGTTGATCT	TCCTCCTTGATTGCTGCTT	81	60
TRINITY_DN23381_c0_g1	GCACAGAATATGTGCCAACT	CTCAACCTCTTGCATGATTCC	100	60
TRINITY_DN9042_c0_g2	AGGGCTTGGAGGAAGTAG	AGAAGGACTCCTCGAAGAT	106	60
TRINITY_DN11178_c0_g1	CGCTGATATATTAGCCCTCG	GGAGCTGGTAGGTTTATGG	129	60

**Table 6 plants-13-03495-t006:** Liquid chromatography elution gradient.

Time (min)	Velocity of Flow (μL min^−1^)	A%	B%
0.0	400	98	2
0.25	400	98	2
10.0	400	2	98
13.0	400	2	98
13.1	400	98	2
15.0	400	98	2

## Data Availability

The data presented in this study are available upon request from the corresponding author.

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
