# Peer review of "Effects of Far-Red Light and Ultraviolet Light-A on Growth, Photosynthesis, Transcriptome, and Metabolome of Mint (Mentha haplocalyx Briq.)"

_plants, 2024, doi:10.3390/plants13243495_

Round 1

Reviewer 1 Report

Comments and Suggestions for Authors

The work interrogates the impact of supplement FR and UVA irradiation on mint plants and presents a comparative transcriptomic and metabolomics analysis.  Multiple features were found to be altered, including transcripts of phytohormone signaling, metabolites such as phytohormones and secondary metabolites, and photosynthesis parameters. Transcriptomic data were validated using three genes. The manuscript is well-organized and is of great interest to the scientific community. The manuscript needs to improve the processing and, if possible, deposit the raw data in a public repository for transcriptomic and metabolomic data. The following are the specific comments.

Name chlorophyll b chlorophyll b properly along the manuscript, for instance in line 54.

Line 57-59. In the last paragraph, evidence for two species is presented, but they are not comparable to each other. In other work, FR was used for grapes and UVB was used for soybeans. It is suggested to provide more examples to back up the conclusion made.

Line 71-73. It would be valuable to give examples of crops that reveal physiological, morphological, and biochemical traits that have been reported as being impacted by LED.

Lines 76-79. References are required.

Lines 83-84. References are required.

Line 95. Clarification that FR means RB + FR is required.

Lines 104-106. Clarifying why UVA inhibits plant height. It seems that UVA does not affect height instead of inhibiting, but may there is data of plants with FR+ UVA?

Lines 117. Adding statistical significance to panels A and B.

Lines 188. Clarification in the text is required as there are DEGs for two groups in a comparative analysis.

Line 193. Double check the redaction of the passage.

Line 195-198. Is there a reason to consider a p-value above 0.05 in the analysis? I am curious as to how realistic comparisons between groups with different p-values are. Would it be better to make comparisons with similar p-values? In this case, FR and UVA were compared to CK.

Section 2.5.1. While FC and VIP facilitate the association of metabolites to groups in comparative analysis, they do not support or validate the identity of metabolites. 1) Clarification on whether MS/MS data were acquired? 2) What are the ppm values for parent ions detected? 3) Defining the level of identification should be valuable, for instance based on analyte, parent ion, or MS/MS.

Table 2. Explain why both comparative groups have a similar total number of metabolites. Does this number correspond to a set of metabolites that have been inspected? Is this set of metabolites in the house database?

Lines 225-227. Briefly describe how the compound ID for KEGG KO was obtained.

Line 241. Clarification is required in the text.  The top ten DAMs are displayed instead of the whole DAMs.

Lines 242-243. Why is the p-value of enrichment different in group comparisons? It is usually accepted if the p-value is below 0.05. Is the analysis reported here affected by p-values above 0.05?

Line 286. is this acronym correct?

Fig 14. Displaying the data in fold change should be valuable for direct comparison with the RN-seq data. If possible, please present the fold change found in RNA-seq following the qRT-PCR data obtained.

Materials and methods section. 1) Define acronyms like EC (line 520) as early as possible in this section. 2) The provider of the instruments used in this magazine format is required.

Table 4. Why is the use of units μmol s-1 instead of μmol m-2 s-1

Line 547. Adding the formula used for calculation would be valuable.

Line 571. A software repository should be valuable.

Line 579. Correcting units

Lines 585-597. 1) parameters for processing data required 2) define the samples and treatments analyzed by qPCR

Section 4.6. 1) define the identity of the internal standard 2) which volume is appropriate? The volume for reconstitution is based on what? 3) I suggest LC-MS instead of LC/MS 4) Does the mass spectrometer utilize ESI? Please clarify in the text 5) line 610 correcting units 6) specified vendors for solvents. Since positive and negative ion modes utilize the same mobile phase combination, these two passages should be combined. 7 ) Data acquisition is required, including gradients for eluded metabolites, among other parameters. Was UV absorption recorded? Is MS/MS data acquired? What is the resolution used for MS? 8) Please define parameters for data processing. 9) Depositing the raw data of a public repository is highly recommended.

I recommend not using expressions in metabolites (lines 289, 297-299)

Improve the resolution of figures 3, 10, 17b

Typos line 103

Author Response

Responses to Reviewer 1 Comments:

Point 1:

Open Review

Quality of English Language

(x) The quality of English does not limit my understanding of the research.
( ) The English could be improved to more clearly express the research.

Yes

Can be improved

Must be improved

Not applicable

Does the introduction provide sufficient background and include all relevant references?

( )

(x)

( )

( )

Is the research design appropriate?

(x)

( )

( )

( )

Are the methods adequately described?

( )

( )

(x)

( )

Are the results clearly presented?

( )

(x)

( )

( )

Are the conclusions supported by the results?

(x)

( )

( )

( )

Comments and Suggestions for Authors

The work interrogates the impact of supplement FR and UVA irradiation on mint plants and presents a comparative transcriptomic and metabolomics analysis. Multiple features were found to be altered, including transcripts of phytohormone signaling, metabolites such as phytohormones and secondary metabolites, and photosynthesis parameters. Transcriptomic data were validated using three genes. The manuscript is well-organized and is of great interest to the scientific community. The manuscript needs to improve the processing and, if possible, deposit the raw data in a public repository for transcriptomic and metabolomic data. The following are the specific comments.

Response 1: We thank this reviewer for his/her positive review on our manuscript. These opinions help to improve academic rigor of our article. Based on your suggestion and request, we have made corrected modifications on the revised manuscript. We hope that our work can be improved again. As your suggestions, the transcriptomic raw data has been uploaded to https://www.ncbi.nlm.nih.gov/. The raw data of the metabolomic has been uploaded to https://www.cncb.ac.cn/.

Point 2: Name chlorophyll b chlorophyll b properly along the manuscript, for instance in line 54.

Response 2: Thank you. We have corrected to make the word harmonized within the whole manuscript. (Lines 59-60)

Point 3: Line 57-59. In the last paragraph, evidence for two species is presented, but they are not comparable to each other. In other work, FR was used for grapes and UVB was used for soybeans. It is suggested to provide more examples to back up the conclusion made.

Response 3: Thank you for your suggestion. We cited more references in the passage to support the proposed conclusion. (Kong et al. discovered that supplementary FR light enhanced the stem length of grapes and increased carbohydrate levels in several organs and maximum net photosynthetic rate in leaves. Three types of UV differentially promote the expression of shikimate pathway genes and production of anthocyanins in grape berries. Ultraviolet-A (UVA) radiation significantly increased the shoot dry weight, leaf area, anthocyanin, and ascorbic acid levels of lettuce. Yang et al. found that compared with the treatment without FR, adding a low or high proportion of FR can increase plant height, biomass, leaf area, and starch content of soybean. Field supplementation of ultra-violet-B radiation (UVB) reduced plant height, leaf area, and biomass of soybean, alongside Pn, transpiration rate (Tr), and water use efficiency (WUE), while enhancing stomatal resistance.) (Lines 49-59)

Point 4: Line 71-73. It would be valuable to give examples of crops that reveal physiological, morphological, and biochemical traits that have been reported as being impacted by LED.

Response 4: Thanks for the suggestion. We cited more examples and revised it to: “The current research on the impact of LEDs on plants mostly concentrates on crops including vegetables and fruits, researching yield, physiological morphology, biochemical, and related factors. Indeed, plants respond through specific photomorphogenic and physiological processes, at both the micro and macro level, such as improvement of growth and photosynthetic capacity in basil, flowering and carbohydrate accumulation of ageratum and salvia, and secondary metabolite content of Myrtus communis L.” (Lines 76-81)

Point 5: Lines 76-79. References are required.

Response 5: According to your suggestion, we have added the reference in the corresponding positions of the manuscript. (Line 88)

Point 6: Lines 83-84. References are required.

Response 6: We have added the reference to the manuscript accordingly. (Line 93)

Point 7: Line 95. Clarification that FR means RB + FR is required.

Response 7: According to your suggestion, we clarified it in this revision. Correspondingly, we have also clarified CK and UVA in the manuscript. “Mint plants under 70% red light and 30% blue light (7R3B) + FR (FR) treatment exhibited greater growth, as evidenced by higher plant height and larger shoots compared to the other two treatments. Plant growth under both 7R3B (CK) and 7R3B + UVA (UVA) treatments were comparatively uniform (Fig. 1).” (Lines 104, 106).

Point 8: Lines 104-106. Clarifying why UVA inhibits plant height. It seems that UVA does not affect height instead of inhibiting, but may there is data of plants with FR+ UVA?

Response 8: UVA had no significant effect on plant height, we have made corrections in the manuscript. In addition, we did not design FR+UVA treatment during the experiment. (Lines 115-116)

Point 9: Lines 117. Adding statistical significance to panels A and B.

Response 9: Thanks. We have added statistical significance to panels A and B. (Line 129)

Point 10: Lines 188. Clarification in the text is required as there are DEGs for two groups in a comparative analysis.

Response 10: Thank you for your kind suggestion. We revised it to: “(B) Number of DEGs detected in FR vs. CK and UVA vs. CK. (C) Venn diagram of DEGs in FR vs. CK and UVA vs. CK.” (Lines 208-209)

Point 11: Line 193. Double check the redaction of the passage.

Response 11: We have reorganized and rewritten the passage as: Pathway enrichment analysis utilized the KEGG Pathway Database of DEGs. Figure 8A and B illustrated the top 20 pathways which were enriched with DEGs. DEGs in FR vs. CK and UVA vs. CK were enriched in pentose and glucuronate interconversions. In addition, DEGs in UVA vs. CK were also enriched in phenylpropanoid biosynthesis, as well as starch and sucrose metabolism. The results demonstrated that FR and UVA light showed intricate influences on biological pathways in mint. (Lines 214-223)

Point 12: Line 195-198. Is there a reason to consider a p-value above 0.05 in the analysis? I am curious as to how realistic comparisons between groups with different p-values are. Would it be better to make comparisons with similar p-values? In this case, FR and UVA were compared to CK.

Response 12: Pathways with p-value > 0.05 are not statistically significant, but this does not mean that they are biologically unimportant. Since these are the top 20 enriched pathways, we believe that these pathways may be altered after FR and UVA treatment, and therefore analyzed them. The number of genes contained in each pathway is also different, and p-value will also be affected. The enrichment results between the differentially expressed genes compared in each group are independent, and the significance is also determined based on the situation of differentially expressed genes between each group, which is different. Based on your suggestion, we revised it to: “Figure 8A and B illustrated the top 20 pathways which were enriched with DEGs. DEGs in FR vs. CK and UVA vs. CK were enriched in pentose and glucuronate interconversions. In addition, DEGs in UVA vs. CK were also enriched in phenylpropanoid biosynthesis, as well as starch and sucrose metabolism”. We hope that these responses are adequate to address your concerns. (Lines 216-221)

Point 13: Section 2.5.1. While FC and VIP facilitate the association of metabolites to groups in comparative analysis, they do not support or validate the identity of metabolites. 1) Clarification on whether MS/MS data were acquired? 2) What are the ppm values for parent ions detected? 3) Defining the level of identification should be valuable, for instance based on analyte, parent ion, or MS/MS.

Response 13: Thanks for your comment. Our metabolomics experiment was conducted by Biomarker Technologies, so we have relevant data for metabolite qualitative and quantitative analysis, data quality assessment, annotation analysis, differential expression analysis, and functional enrichment. Based on your suggestion, we have added the following explanations in the manuscript: The raw data collected using MassLynx V4.2 is processed by Progenesis QI software for peak extraction, peak alignment and other data processing operations, based on the Progenesis QI software online METLIN database and Biomark’s self-built library for identification, and at the same time, theoretical fragment identification and parent ion mass deviation all are within 100ppm. (Lines 233-237)

Point 14: Table 2. Explain why both comparative groups have a similar total number of metabolites. Does this number correspond to a set of metabolites that have been inspected? Is this set of metabolites in the house database?

Response 14: The total number is the total number of metabolites identified in the three treatments, which is referenced from metlin public databases and Biomaker’s self-built library. We deleted it from the table.

Point 15: Lines 225-227. Briefly describe how the compound ID for KEGG KO was obtained.

Response 15: We can access it from the KEGG pathway official website (https://www.kegg.jp/) obtain the compound ID for KEGG KO. We have added it to the manuscript. (Lines 258-260)

Point 16: Line 241. Clarification is required in the text. The top ten DAMs are displayed instead of the whole DAMs.

Response 16: Thank you for your kind suggestion. We revised it to: “(B) The top ten up and down -regulated DAMs of fold change between FR and CK treatments. (C) The top ten up and down -regulated DAMs of fold change between UVA and CK treatments.” (Lines 277-278)

Point 17: Lines 242-243. Why is the p-value of enrichment different in group comparisons? It is usually accepted if the p-value is below 0.05. Is the analysis reported here affected by p-values above 0.05?

Response 17: The number of genes contained in each pathway is different, and the p-value will also be affected. Its significance is also determined based on the differentially expressed genes in each group, so comparing different p-values between groups is also meaningful. The pathways with p-value above 0.05 is not statistically significant, but this does not mean that they are biologically unimportant. Since these are the top 20 enriched pathways, we believe that these pathways may undergo changes after FR and UVA treatment. And our sequencing was analyzed uniformly after being tested by a professional company, so we conducted an analysis on it. According to your suggestion, we have revised it to: “The KEGG pathways of DAMs in the FR vs. CK group were enriched in pyruvate metabolism, monobactam biosynthesis, and isoquinoline alkaloid biosynthesis. Furthermore, DAMs in the UVA vs. CK group exhibited enrichment in porphyrin metabolism and glutathione metabolism (Fig. 10D, E)”. (Lines 268-273)

Point 18: Line 286. is this acronym correct?

Response 18: Yes. The gene acronyms in the manuscript are uniformly referenced from the KEGG ORTHOLOGY database.

Point 19: Fig 14. Displaying the data in fold change should be valuable for direct comparison with the RN-seq data. If possible, please present the fold change found in RNA-seq following the qRT-PCR data obtained.

Response 19: Based on your suggestion, we analyzed foldchange and found the upregulation trend of AUX/IAA, DELLA, and POD was consistent between RNA-seq and qRT-PCR (Fig 14B). In addition, we increased the FPKM expression level of RNA seq in Figure 14A. And the combination of these three factors can made our RNA-seq data more rigorous. (Line 365)

Fig: Please see the attachment.

Point 20: Materials and methods section. 1) Define acronyms like EC (line 520) as early as possible in this section. 2) The provider of the instruments used in this magazine format is required.

Response 20: Thanks for the suggestion. 1) We have revised it to: “electrical conductivity (EC)”. (Line 571) 2) We have checked the manuscript and added the provider of the instruments used.

Point 21: Table 4. Why is the use of units μmol s-1 instead of μmol m-2 s-1

Response 21: PPF is the acronym for Photosynthetic Photon Flux, with the unit of μmol s-1. It represents how many photosynthetic photons a plant growth lamp can emit in one second. When designing and formulating the spectral form of a plant lamp, PPF has already been determined. PPFD stands for Photosynthetic Photon Flux Density. The unit is μmol m-2 s-1, which represents the PPF within one square meter. PPFD measures the number of photons falling on plants. In addition to the influence of their own lighting fixtures, the height and surface reflectivity of plants and lighting fixtures also affect the PPFD value. In short, PPF is a parameter set during the design of the lamp tube, while PPFD is a parameter that is illuminated onto the plant during actual use. So we presented both parameters in the materials and methods section.

Point 22: Line 547. Adding the formula used for calculation would be valuable.

Response 22: Thanks for the suggestion. We have added the formula in the manuscript:

Formula: Please see the attachment.

In the formula, Ca and Cb represent the concentrations of chlorophyll a and chlorophyll b, respectively; Cx c is the total concentration of carotenoids; A665, A649, and A470 represent the absorbance of chloroplast pigment extract at wavelengths of 665nm, 649nm, and 470nm, respectively. (Lines 602-610)

Point 23: Line 571. A software repository should be valuable.

Response 23: Thank you for your kind suggestion. Due to the fact that the curve fitting software we are using is a web version, we have attached the website URL at the corresponding position in the manuscript: https://photosynthetic.sinaapp.com. Hoping it will be valuable. (Lines 634-635)

Point 24: Line 579. Correcting units

Response 24: We revised it to: “ng μl-1”. (Line 645)

Point 25: Lines 585-597. 1) parameters for processing data required 2) define the samples and treatments analyzed by qPCR

Response 25: Thank you for pointing this out. 1) We have added specific parameters and clarification in the manuscript as follows: Real-time quantitative RT-PCR: Quantification was performed with a two-step reaction process: reverse transcription (RT) and PCR. Each RT reaction consisted of 0.5 μg RNA, 2 μl of 5×TransScript All-in-one SuperMix for qPCR and 0.5 μl of gDNA Remover, in a total volume of 10 μl. RT Reactions were performed in a GeneAmp® PCR System 9700 (Applied Biosystems, USA) for 15 min at 42℃,5 s at 85℃.The 10 μl RT re-action mix was then diluted × 10 in nuclease-free water and held at -20℃. Real-time PCR was performed using LightCycler® 480 Ⅱ Real-time PCR Instrument (Roche, Swiss) with 10 μl PCR reaction mixture that included 1 μl of cDNA, 5 μl of 2×PerfectStartTM Green qPCR SuperMix, 0.2 μl of forward primer, 0.2 μl of reverse primer and 3.6 μl of nuclease-free water. Reactions were incubated in a 384-well optical plate (Roche, Swiss) at 94℃ for 30 s, followed by 45 cycles of 94℃ for 5 s, 60℃ for 30 s. Each sample was run in triplicate for analysis. At the end of the PCR cycles, melting curve analysis was performed to validate the specific generation of the expected PCR product. 2) The expression levels of AUX/IAA (TRINITY_DN23381_c0_g1), DELLA (TRINITY_DN9042_c0_g2), and POD (TRINITY_DN11178_c0_g1) genes were analyzed using qRT-PCR under CK, FR, and UVA treatments, to verify the reliability of the RNA-Seq data. (Lines 653-673)

Point 26: Section 4.6. 1) define the identity of the internal standard 2) which volume is appropriate? The volume for reconstitution is based on what? 3) I suggest LC-MS instead of LC/MS 4) Does the mass spectrometer utilize ESI? Please clarify in the text 5) line 610 correcting units 6) specified vendors for solvents. Since positive and negative ion modes utilize the same mobile phase combination, these two passages should be combined. 7) Data acquisition is required, including gradients for eluded metabolites, among other parameters. Was UV absorption recorded? Is MS/MS data acquired? What is the resolution used for MS? 8) Please define parameters for data processing. 9) Depositing the raw data of a public repository is highly recommended.

Response 26: Thank you for your comment. We have made additional supplements to section 4.6: Sample extraction: 1. Take out the sample stored at -80℃, thaw it at room temperature, weigh 50 mg of the sample and add 1000 μl of extraction solution containing internal standards (methanol: acetonitrile: water = 2:2:1, internal standard is 2-Chloro-L-phenylalanine, internal standard concentration 20 mg L-1). Vortex mix for 30 s. 2. Add steel beads and process in a grinder at 45 Hz for 10 min, followed by sonication for 10 min in an ice-water bath. 3. Stand at minus 20℃ for 1 h. 4. Centrifuge the sample at 4℃ and 12000 rpm for 15 min. 5. Carefully remove 500 μl supernatant into the EP tube. 6. Dry the extract in a vacuum concentrator. 7. Add 160 μl of extraction solution (acetonitrile: water=1:1) to the dried metabolites for reconstitution (in order to ensure that the QC sample volume meets the injection requirements after configuration). 8. Vortex for 30 s and sonicate in an ice-water bath for 10 min. 9. Centrifuge the sample at 4℃ and 12000 rpm for 15 min. 10. Carefully take out 120 μl supernatant in 2 ml injection bottle, and take 10 μl of each sample and mix it into QC sample for machine testing. (1. Methanol CAS: 67-56-1; pureness: LC-MS; Merck, Germany. 2. Acetonitrile CAS: 75-05-8; pureness: LC-MS; Merck, Germany; 3. 2-Chloro-L-phenylalanine CAS: 64-18-6; pureness: LC-MS; Aladdin, China.)

The LC-MS system for metabolomics analysis is composed of Waters Acquity I-Class PLUS ultra-high performance liquid tandem Waters Xevo G2-XS QTof (Waters, USA) high resolution mass spectrometer. The column used is purchased from Waters Acquity UPLC HSS T3 column (1.8 μm 2.1*100 mm) (Waters, USA). Positive ion mode: mobile phase A: 0.1% formic acid aqueous solution; mobile phase B: 0.1% formic acid acetonitrile. Negative ion mode: mobile phase A: 0.1% formic acid aqueous solution; mobile phase B: 0.1% formic acid acetonitrile. The elution gradients are shown in Table 6. Column temperature 50℃, injection volume 1 μl. (Formic acid CAS: 64-18-6; pureness: LC-MS; TCI, USA.)

Waters Xevo G2-XS QTOF high resolution mass spectrometer can collect primary and secondary mass spectrometry data in MSe mode under the control of the acquisition software (MassLynx V4.2, Waters, USA). In each data acquisition cycle, dual-channel data acquisition can be performed on both low collision energy and high collision energy at the same time. The low collision energy is 2V, the high collision energy range is 10~40V, and the scanning frequency is 0.2 s for a mass spectrum. The parameters of the ESI ion source are as follows: Capillary voltage: 2000V (positive ion mode) or -1500V (negative ion mode); cone voltage: 30V; ion source temperature: 150°C; desolvent gas temperature 500°C; backflush gas flow rate: 50L h-1; Desolventizing gas flow rate: 800L h-1; Collection range of mass charge ratio (m/z): 50-1200. (No UV absorption, as there is no corresponding analyzer in series, MS resolution is 25000-35000).

The raw data collected using MassLynx V4.2 (Waters, USA) is processed by Progenesis QI software for peak extraction, peak alignment and other data processing operations, identification based on the Progenesis QI software online METLIN database and Biomark’s self-built library, and at the same time, theoretical fragment identification and mass deviation are all within 100 ppm.

After normalizing the original peak area information with the total peak area, the follow-up analysis was performed. Principal component analysis and Spearman correlation analysis were used to judge the repeatability of the samples within group and the quanlity control samples. The identified compounds are searched for classification and pathway information in KEGG, HMDB and lipidmaps databases. According to the grouping information, calculate and compare the difference multiples, T test was used to calculate the difference significance p value of each compound. The R language package ropls was used to perform OPLS-DA modeling, and 200 times permutation tests was performed to verify the reliability of the model. The VIP value of the model was calculated using multiple cross-validation. The method of combining the difference multiple, the P value and the VIP value of the OPLS-DA model was adopted to screen the differential metabolites. The screening criteria are FC>1, P value<0.05 and VIP>1. The difference metabolites of KEGG pathway enrichment significance were calculated using hypergeometric distribution test. (Lines 680-737)

Table 6 Liquid chromatography elution gradient

Time (min)

Velocity of flow (μl min-1)

A%

B%

0.0

400

98

2

0.25

400

98

2

10.0

400

2

98

13.0

400

2

98

13.1

400

98

2

15.0

400

98

2

Point 27: I recommend not using expressions in metabolites (lines 289, 297-299)

Response 27: According to your suggestion, we avoided using expression to describe metabolites, and we rephrased this section as follows: Anethole, coniferin, and syringin were down-regulated in UVA (Fig. 12). (Lines 327-328) …… In FR treatment, the up-regulated metabolite included 2',3,4,4',6'-pentahydroxychalcone. Conversely, the down-regulated metabolites encompassed butein, phlorizin, and naringenin. Pinobanksin 3-acetate was down-regulated in response to UVA exposure (Fig 13). (Lines 335-338)

Point 28: Improve the resolution of figures 3, 10, 17b

Response 28: Thanks. We have revised the figure 3, 10, 17b.

Point 29: Typos line 103

Response 29: Thank you for your suggestion. After careful verification, we found that an additional period was entered before the parentheses. We have removed it. (Line 113)

Reviewer 2 Report

Comments and Suggestions for Authors

This interesting study deals with the effects of light quality in terms of its spectral composition on the growth performance of mint (Mentha haplocalyx). the investigation was performed at morphological, physiological, metabolomic and transcriptomic level. Although the subject is not original this is the first time these aspects are studied in mint. Results have practical implications for the cultivation of mint in controlled conditions.

Overall the article is well structured. The experimental design is appropriate and the results are clearly presented. My only concerns are as it follows:

- The Title: can be modified as suggested (do not use acronims in the title and the term 'treatments' can be omitted).

- Line 52 Use extended terms before introducing a new acronym.

- Lines 107-108 This sentence of the section Results is not clear to me; please rephrase it.

- Line 141 This sentence is not clear to me: please rephrase it.

- Line 460 In the Discussion cite the role of Phenylalanine ammonia-lyase in plant responses to biotic and abiotic stresses; I indicated two relevant references pertaining this aspect.

- Line 621 Was the number of replicates variable in each experiment or for each treatment (minimum three) or was it always three? Please confirm. The same questions applies to the captions of Table 1 and Figures 2,3. 5-7, 8, 10-14 (see notes in the text, attached PDF file), please change accordingly.

For additional very minor text editings see notes in the text (attached PDF file).

Author Response

Responses to Reviewer 2 Comments:

Point 1:

Open Review

Quality of English Language

(x) The quality of English does not limit my understanding of the research.
( ) The English could be improved to more clearly express the research.

Yes

Can be improved

Must be improved

Not applicable

Does the introduction provide sufficient background and include all relevant references?

(x)

( )

( )

( )

Is the research design appropriate?

(x)

( )

( )

( )

Are the methods adequately described?

( )

(x)

( )

( )

Are the results clearly presented?

( )

(x)

( )

( )

Are the conclusions supported by the results?

( )

(x)

( )

( )

Comments and Suggestions for Authors

This interesting study deals with the effects of light quality in terms of its spectral composition on the growth performance of mint (Mentha haplocalyx). the investigation was performed at morphological, physiological, metabolomic and transcriptomic level. Although the subject is not original this is the first time these aspects are studied in mint. Results have practical implications for the cultivation of mint in controlled conditions.

Overall the article is well structured. The experimental design is appropriate and the results are clearly presented. My only concerns are as it follows:

Response 1: We feel great thanks for your professional and positive review on our manuscript.

Point 2: The Title: can be modified as suggested (do not use acronims in the title and the term 'treatments' can be omitted).

Response 2: Thank you for your kind suggestion. We have modified it to: Effects of Far-Red Light and Ultraviolet Light-A on Growth, Photosynthesis, Transcriptome, and Metabolome of Mint (Mentha haplocalyx Briq.).

Point 3: Line 52 Use extended terms before introducing a new acronym.

Response 3: Thanks for the suggestion. We revised it to: “ultraviolet-B radiation (UVB)”. (Line 57)

Point 4: Lines 107-108 This sentence of the section Results is not clear to me; please rephrase it.

Response 4: We have rewritten this sentence as follows: Compared with CK, the plant width increased by 13.07% under FR treatment, but there was no significant difference between UVA and CK. (Lines 117-119)

Point 5: Line 141 This sentence is not clear to me: please rephrase it.

Response 5: We have rephrased it to: In contrast with CK treatment, there was no significant difference in the Pn of FR and UVA treatments but the trend of Pn under the three treatments was shown as: FR > CK > UVA. (Line 158)

Point 6: Line 460 In the Discussion cite the role of Phenylalanine ammonia-lyase in plant responses to biotic and abiotic stresses; I indicated two relevant references pertaining this aspect.

Response 6: We sincerely appreciate your valuable comments. According to your suggestion, we have added the two references you recommended to support the relevant conclusions of phenylalanine ammonia-lyase, and they are referred as [54] and [57].

Point 7: Line 621 Was the number of replicates variable in each experiment or for each treatment (minimum three) or was it always three? Please confirm. The same questions applies to the captions of Table 1 and Figures 2,3. 5-7, 8, 10-14 (see notes in the text, attached PDF file), please change accordingly.

Response 7: Our experiments were replicated always three times. Correspondingly, revisions have been made on Table 1 and Figures 2, 3. 5-7, 8, 10-14.

Point 8: For additional very minor text editings see notes in the text (attached PDF file).

Response 8: Thanks for your careful checks. Based on your comments, we have made the corrections one by one in the manuscript.

Round 2

Reviewer 1 Report

Comments and Suggestions for Authors

All comments have been addressed properly. No new comment has arisen in the review.